# Improved Confidence-Interval Estimations Using Uncertainty Measure and Weighted Feature Decisions for Cuff-Less Blood-Pressure Measurements

**DOI:** 10.3390/bioengineering12020131

**Published:** 2025-01-30

**Authors:** Soojeong Lee, Mugahed A. Al-antari, Gyanendra Prasad Joshi

**Affiliations:** 1Department of Computer Engineering, Sejong University, 209 Neungdong-ro, Gwangjin-gu, Seoul 05006, Republic of Korea; leesoo86@sejong.ac.kr; 2Department of Artificial Intelligence and Data Science, Sejong University, Seoul 05006, Republic of Korea; 3Department of AI Software, Kangwon National University, Kangwon State, Samcheok 10587, Republic of Korea

**Keywords:** uncertainty, confidence intervals, cuff-less blood-pressure estimation, Gaussian processes regression, gradient boosting algorithm

## Abstract

This paper presents a method to improve confidence-interval (CI) estimation using individual uncertainty measures and weighted feature decisions for cuff-less blood-pressure (BP) measurement. We obtained uncertainty using Gaussian process regression (GPR). The CI obtained from the GPR model is computed using the distribution of BP estimates, which provides relatively wide CIs. Thus, we proposed a method to obtain improved CIs for individual subjects by applying bootstrap and uncertainty methods using the cuff-less BP estimates of each subject obtained through GPR. This study also introduced a novel method to estimate cuff-less BP with high fidelity by determining highly weighted features using weighted feature decisions. The standard deviation of the proposed method’s mean error is 2.94 mmHg and 1.50 mmHg for systolic blood pressure (SBP) and (DBP), respectively. The mean absolute error results were obtained by weighted feature determination combining GPR and gradient boosting algorithms (GBA) for SBP (1.46 mmHg) and DBP (0.69 mmHg). The study confirmed that the BP estimates were within the CI based on the test samples of almost all subjects. The weighted feature decisions combining GPR and GBA were more accurate and reliable for cuff-less BP estimation.

## 1. Introduction

Most elderly people and patients using home blood-pressure (BP) monitoring devices measure their BP only twice daily due to the inconvenience of conventional cuff-based monitors. Thus, there is an urgent need for cuff-less BP monitoring devices that can effectively cope with dynamic BP changes and improve accuracy and reliability. These devices are essential for intensive critical care. However, accurate BP measurement is a challenge due to the significant impact of the inherent physiological variability of BP. This variability, influenced by stress, diet, exercise, climate, and disease, continuously fluctuates, making estimation more intricate. Specifically, physical activity and psychological stress affected BP variability during the 5 min before BP measurement. Psychological stress, expressed as negative emotions and workplace location, increases more significantly with physical activity than psychological stress. Stressful situations were significantly associated with BP elevation [1]. Schwartz et al. [2] reported that body position and location during measurement accounted for a significant portion of BP variability. Therefore, cuff-less BP measurement devices require a method to measure the uncertainty that may arise from physiological variability. This is a significant limitation since most devices provide single-point estimates without confidence intervals (CIs).

Cuff-less BP measurements are naturally affected by various sources of uncertainty, leading to discrepancies between the measured value (the estimate) and the actual BP value (the reference BP) [3]. The sources of uncertainty can be categorized into random and systematic errors, which will be discussed in detail in the section on uncertainty measurement. When multiple sources of uncertainty influence a cuff-less BP measurement, the distribution of these BP measurements may converge to a Gaussian distribution as the number of uncertainties increases, regardless of the original distribution of the parameters representing these uncertainties [4]. Although several researchers in this field have attempted to study the uncertainty of biosignal measurements [5], an attempt has not been made to integrate the quality characteristics of the obtained signal and its compatibility with the estimation algorithm into an overall confidence measure of measurement accuracy.

Therefore, it is necessary to provide CI for cuff-less BP measurement to assess uncertainty, and it provides various estimated BP values that may include significant unknown factors [3]. If the integrated statistics show a wide range of CI, it can be considered a warning sign and alert patients, healthcare providers, and families to the risk. However, CI estimation in cuff-less BP measurement is still in the early stages of research and has not been actively studied. In addition, estimating the CI for each patient in practice requires multiple repeated BP measurements, even with a cuff-less BP monitor. Unfortunately, measuring BP multiple times over an extended period for each patient using a cuff-less BP device is expensive and difficult to achieve because it does not ensure consistent conditions for reproducible readings [6]. Recently, Lee et al. [7] proposed a machine learning (ML) approach to simultaneously estimate BP and CI for cuff-less BP measurements via hybrid feature selection based on photoplethysmography (PPG) with electrocardiogram (ECG). However, this method results in a significant mean absolute error, which results in a CI that is too wide. If the CI is too broad, a warning may recommend discarding the measurement and starting a new one.

Recently, Huang et al. [8] introduced cuff-less BP monitoring using a transformer encoder and stacked attention-gated recurrent units. Sideris et al. [9] proposed post-filtering steps based on long short-term memory (LSTM) using the BP signal quality. Qiu et al. [10] introduced a novel technique to predict BP by combining window functions and piecewise neural networks (NNs) based on the PPG signals. They weighted features from the extracted features and used them as features for BP estimation. Determining weighted features (selecting high-weighted features, excluding similar and low-weighted features [11]) is essential for improving the performance of ML algorithms. Yang et al. [12] developed neighborhood component analysis (NCA) to select weighted feature vectors with high computational speed, but it may exclude valuable features. In a previous study, robust neighborhood component analysis (RNCA) was developed to improve the performance of neighborhood component analysis [13]. The minimum redundancy maximum relevance (MRMR) algorithm can select features quickly; however, they were limited by missing valid features when estimating response variables [14]. Developers or users also need to experimentally determine the number or weights of features.

Our study aims to improve CI estimation using individual uncertainty measurement and weighted feature decisions (WFD) for cuff-less BP measurements. Thus, we use the Gaussian process regression (GPR) [15] to obtain an uncertainty. This uncertainty cannot be obtained directly from NN [16], SVM [17], or deep neural network (DNN) [18]. However, the CIs obtained using the GPR model are calculated based on the distribution of BP estimates, which provides a relatively wide CI. Although the probability that the BP estimate is included in relatively wide CIs increases, it has the disadvantage of reducing the reliability of the BP monitoring system. In addition, if the CI is too narrow, even a tiny change in the BP estimate can easily lead to an out-of-CI, which limits its role as a BP monitoring system. Securing an appropriate range of CIs is necessary to overcome these shortcomings.

To address the above limitations, we propose a method to obtain improved CI for individual subjects by applying bootstrap [19] and uncertainty [20] based on the cuff-less BP estimates of each subject obtained through GPR. Second, this study introduces a novel methodology to estimate cuff-less BP with high fidelity by determining high-weighted features using weighted feature decisions (WFD) because weighted feature extraction is one of the fundamental steps in ML. The WFD method is an algorithm that automatically selects highly weighted features using a unified feature set. In the WFD methodology, we combine MRMR with the gradient boosting algorithm (GBA) to determine the feature set for the best BP estimation results. The role of WFD is to select the weighted feature subset with the smallest mean square error using GBA. In detail, features are received from the integrated feature set, selected, and output as the weighted feature subset for the MRMR algorithm. The weighted feature sets are inputs to the WFD methodology, which selects the optimal feature combination. This paper has the following contributions. Figure 1 is a block diagram of the proposed methodologies.

We develop a method to combine dual-step preprocessing and multiple feature extraction (DPFE) and WFD based on GPR to improve the reliability of cuff-less BP estimates and individual uncertainty estimates.The proposed method measures uncertainties such as CIs and expanded uncertainty for cuff-less BP measurement.The proposed method provides more accurate prediction performance and uncertainty by providing lower standard deviation (SD), mean absolute error, and CI. Therefore, it can help estimate BP and CI for clinical practice or diagnostic purposes.

The organization of the rest of the text on the proposed framework for cuff-less BP and uncertainty estimations is as follows: Section 2 discusses the collection of a dataset containing PPG, ECG, and arterial BP signals; Section 3.1 presents the dual-step preprocessing; Section 3.2 denotes the signal quality index; Section 3.3 discusses the review of feature extraction; the MRMR is presented in Section 3.4; Section 3.5 presents the GBA; Section 4 denotes the GPR for BP and uncertainty estimation; Section 5 denotes the uncertainty estimation; Section 6 provides the experimental results; Finally, Section 7 and Section 8 present the discussion and conclusions, respectively.

## 2. Data Collection

We utilized the multiparameter intelligent monitoring (MIMIC-II) databases Goldberger et al. [21] at the ML storage center of the University of California, Irvine. We acquired 3000 records (participants) of ECG, finger PPG, and arterial BP signals at a sampling frequency of 125 Hz, sufficient to extract consecutive BP data. We obtained reference systolic blood pressure (SBP) and diastolic blood pressure (DBP) from the arterial BP signals. PPG and ECG signal waveforms were combined to obtain a feature set. We extracted statistical features from the PPG signal waveform and PPG signal frequency domain. The duration of the records in the database varied from 8 s to over 480 s. For consistency, a 20 s segment was extracted after 60 s, resulting in 2500 samples from each record, enhancing the reliability of the obtained patient records. Furthermore, hypertension is classified into three categories based on the recently published MIMIC III BP dataset by Fuadah et al. [22]. BP of 140/80–159/99 mmHg is categorized as Level 1 hypertension according to Whelton et al. [23]. Reboussin et al. defined Level 2 hypertension as 160/100–179/109 mmHg [24]. Finally, Aronow et al. defined a BP reading exceeding 180/120 mmHg as a hypertensive emergency that can lead to death [25]. Hypotension is defined as BP of <90/60 mmHg [22]. Thus, developing a system that can accurately estimate the uncertainty of cuff-less BP monitoring and keep track of rapid BP changes is crucial. We ensured the reliability of the data range by including high and low data and excluding unusual outliers, such as very extreme, very high, or low BP readings (SBP ≥ 200, SBP ≤ 70, DBP ≥ 150, DBP ≤ 40 mmHg).

## 3. Methods

### 3.1. Dual-Stage Preprocessing for Features Extraction

Signal preprocessing removes outliers and extracts valuable features from PPG and ECG signals. In particular, various artifacts often contaminate ECG signals, which can degrade signal quality and hinder accurate analysis. Body movements or sensor displacements cause motion artifacts. These artifacts can cause irregularities in ECG waveforms, especially in walking or moving environments. Nagai et al. [26] introduced an algorithm to remove motion artifacts superimposed on ECG signals using stationary wavelet transform. Gunasekaran et al. [27] proposed a novel method to remove real-time artifacts from ECG signals using recursive independent component analysis (ICA). They describe a systematic preprocessing pipeline that adaptively estimates the mixing and demixing matrices of the ICA model while streaming data are being processed.

In our study, we first removed NaNs from the signals to maintain the alignment of each participant. We then performed a detailed validation using the signal quality index (SQI) algorithm to remove bad segment signals from the PPG and ECG signals, which is further described in the following subsections. Then, before denoising, we used an empirical Bayesian method using Cauchy to remove noise from the ECG and PPG data in low and high-frequency wavelets, discarded the last four detailed wavelet transform coefficients (d7, d8, d9, and d10), and approximated the coefficient a10 to remove artifacts from the signal [28].

Subsequently, we decomposed the PPG and ECG signals using a maximal overlap discrete wavelet transform (MODWT) with a Daubechies 8 wavelet (db8) [28]. We then reconstructed the PPG and ECG signals using an inverse MODWT without the high- and low-frequency coefficients. This step is the first stage of signal preprocessing, as shown in Figure 1b. We also used a high-pass filter to suppress noise, using a frequency threshold of 0.5 Hz in the normalized form (Figure 1b). We specified an infinite impulse response (IIR) peculiar to the filter for noise filtering. We set “steepness” S (=0.5)’ to a scalar with interval values [0.5, 1], specifying the transition band steepness. Increasing the filter response makes an ideal high-pass response achievable and increases the resulting filter length and computational cost of filtering. We chose this noise-cancellation method because of its better phase response to these biosignals, higher computational efficiency, and better adaptability [29]. This step is crucial in preparing the PPG, ECG, and ABP signals for the subsequent processing stage. The preprocessing step segmented each 20-s window of the noise-removed ECG, PPG, and ABP signals into smaller segments of fewer heart cycles, which was the next step in feature extraction. In detail, the 20-s window of the denoised ECG, PPG, and ABP signals were divided into 10 to 13 small segments of 1.4 s each, corresponding to shorter cardiac cycles used in the next feature extraction step. Thus, we performed a four-stage wavelet decomposition of the ECG waveform at the “sym4” wavelet. We reconstructed only the second- and third-order detailed coefficients to obtain the ECG signals that made the R-peak more visible for detection (Figure 1b). This meticulous second-stage preprocessing resulted in 59,097 records ready for feature extraction.

### 3.2. Signal Quality Index (SQI)

We used the SQI method presented in [30], which uses an approach to automatically qualify PPG segments collected from patients using the SQI. The SQI algorithm’s output is in binary format (“trusted” and “untrusted”) to simplify interpretation and ease of use. We also demonstrate the effectiveness of SQI to improve the performance of the proposed algorithm. We also used a three-point peak search method with an empirically determined threshold to find PPG pulse peaks and ECG R-peaks. If more than two PPG pulse peaks and ECG R-peaks were missed or more than two noise peaks were mistakenly identified as PPG pulse peaks and ECG R-peaks in a 20-s segment sample, the heart rate (HR) value ultimately derived from that sample was treated as incorrect. Figure 2 shows the flow chart of the SQI algorithm. The detailed steps are given below.

The first step of the SQI algorithm is to detect PPG pulse peaks and ECG R-peaks in a segment sample and compare the detector output to a set of physiologically relevant rules. The following three rules are applied sequentially, and if none are satisfied, the segment sample is classified as “bad”. Rule 1: The extrapolated HR from a 20 s segment sample should be between 40 and 180 beats per minute (bpm). (There may be HRs outside this range, but this is the physiologically feasible HR range for the adult population that wearable sensors are likely to use.) Rule 2: We allow a maximum interval of 3 s between consecutive PPG pulse peaks or ECG R-peaks. This rule, which ensures that no more than one beat is missed, is designed to guide analysis. Rule 3: We also require that the maximum to minimum beat interval ratio within a segment sample be less than 2.2 s. This is a precise limit, as we expect HR to not change more than 10% in a 20 s segment sample. We use a limit of 2.2 s to allow for one missed beat. If all three rules are met, an adaptive QRS/PPG pulse template matching approach is used, as described below.

Template matching algorithms have been instrumental in identifying ventricular ectopic beats in PPG and ECG [30]. In PPG or ECG segment samples, template matching can determine the regularity of the segments regardless of the actual shape of the PPG pulse waveform or QRS complexes, which serves as an indicator of reliability (since segments contaminated by artifacts have irregular shapes). Our processing sequence is as follows:We compute the median beat interval using all detected R-peaks/PPG pulse peaks in each PPG/ECG sample.We leave no stone unturned in our analysis. Individual PPG pulses and QRS complexes are extracted by taking a window centered around each detected PPG pulse peak and R-peak, with the window width being the median beat interval.We obtain the average PPG pulse template by calculating the average of all PPG pulses in the sample. Then, we compute the correlation coefficient of each PPG pulse and the average PPG pulse template. We also acquired the average QRS template by taking the mean of all QRS complexes in the sample. Then, we calculate the correlation coefficient of individual QRS complexes with the average QRS template.The average correlation coefficient is finally obtained by averaging all correlation coefficients of all PPG and ECG samples. Examples of QRS complex and PPG pulses and generated templates from morphologically regular and irregular signal samples are shown in Figure 2, respectively.

### 3.3. Review of Feature Extraction

Extracting valuable features from PPG with ECG signals is crucial to accurately predict cuff-less BP and cardiac index after initial and secondary preprocessing. Therefore, we investigated the PPG signal’s time and frequency domains, extracted pulse transit time (PTT)-related features using the PPG and ECG signals, and further extracted statistical features by analyzing the shape of the PPG signal waveform. Examining the PPG signal’s frequency domain revealed the power concentration in the low-frequency band. Subsequently, features were extracted using the pulse shape of the PPG signal and the time between ECG and PPG signals as the time axis in the upper panel of Figure 3. Initially, PTT [10] was used to determine the time required for arterial pulse waves to reach the peripheries. The PTT was derived from the R-waves on the PPG and ECG of the oxygen saturation monitor. Consequently, BP can be indirectly calculated by assuming that the PTT wave velocity is inversely proportional to the systolic blood pressure (SBP).

The relationship between PTT and SBP is based on the assumption that as SBP increases, arteries become stiffer (especially in hypertension) [10]. Thus, the wave velocity increases, shortening the PTT. Conversely, as BP decreases and the wave velocity decreases, the arterial wall becomes more flexible, lengthening the PTT. If PTT could be measured using pulse waves based on PPG signals in a wearable device, the time it takes for the pressure wave to travel between two points could be continuously recorded. Therefore, PTT could be measured in an emergency, and the inverse proportionality between PTT and BP could be used to estimate SBP.

In contrast, ECG R-peaks were used to estimate the aortic valve opening and the left ventricle contraction. This allows the estimation of the peripheral pulse rate arrival using PPG, and the PTT can be obtained from both values. We also calculated the pulse arrival time, which represents the time interval between the ECG R-peak and the PPG rise point. The pulse arrival time and PTT are useful features for estimating BP values [10]. Another important feature used was the PPG pulse intensity ratio (PIR), which is inversely proportional to the diastolic trough [10]. The PPG signals allowed us to observe the waveform associated with the heart rate cycle. Therefore, estimating the PPG pulse corresponding to the heart cycle reveals that the rising edge is the systolic time (ST), and the falling edge is the diastolic time (DT) [10]. Furthermore, the regions within the pulse corresponding to the ST and DT served as the systolic area (SA) and the diastolic area (DA), respectively, as shown in Figure 3. The characteristic value of the PPG waveform was calculated as follows:(1)$${ppg}_{p}=PPG\left(p_{t(1)}:p_{t(2)}\right)$$
where ${ppg}_{p}$ is calculated from the PPG’s first trough, $p_{t(1)}$ to PPG’s second trough, $p_{t(2)}$ in the PPG waveform [31]. The negative y-axis values of the PPG waveform are adjusted as in (Equation 2):(2)$${ppg}_{p}={ppg}_{p}-\min\left(ppg_{p}\right)$$(3)$${ppg}_{t}=\frac{p_{t(2)}-p_{t(1)}}{fs}$$
where ${ppg}_{t}$ is the distance from the first $p_{t(1)}$ to the second $p_{t(2)}$ trough in the PPG waveform divided by fs.(4)$$p_{m}=\frac{{ppg}_{area}\times fs^{-1}}{ppg_{t}}$$
where $p_{m}$ denotes the area under one PPG waveform divided by the length of the PPG waveform ${ppg}_{t}$ as in (Equation 3). Here, we use the characteristic parameter of the PPG waveform, as in (Equation 5):(5)$${ppg}_{c}=\frac{p_{m}-p_{d}}{p_{s}-p_{d}}$$
where $p_{s}$ and $p_{d}$ denote the maximum and minimum amplitudes of PPG waveforms, respectively. Furthermore, the PPG pulse areas corresponding to the ST and DT width were identified as the systolic and diastolic areas, respectively. Each PPG pulse is divided into these two areas, which is a major finding, as shown in Figure 3. The characteristic parameters within each PPG pulse, including the ST, DT, cycle time (CT), and width-dependent parameters, were extracted at 10%, 25%, 33%, 50%, 66%, and 75% of the systolic and diastolic areas (SW and DW) [10]. In addition, statistical features such as skewness, kurtosis, and entropy were extracted from the PPG pulse. Table 1 summarizes all the features.

### 3.4. MRMR

The MRMR method selects the most valid set of features that are maximally different from each other and can represent the dependent variable [14]. It works by minimizing redundancy within the feature set and maximizing the relevance of the features to the dependent variable. In other words, the goal of the MRMR algorithm is to identify the optimal subset of features, denoted as *S*, that maximizes $MA_{S}$, representing the relevance of *S* to the predictor value *y*, while minimizing $MI_{S}$, denoting the redundancy of *S*. In this context, $MA_{S}$ and $MI_{S}$ can be expressed using mutual information *I* as follows:(6)$$MA_{S}=\frac{1}{\left|S\right|}\sum_{x\in S}I\left(x,y\right),\;MI_{S}=\frac{1}{{\left|S\right|}^{2}}\sum_{x,z\in S}I\left(x,z\right)$$
where |S| is the number of features in *S*. We think all $2^{\left|\phi \right|}$ combinations, where ϕ is the original feature set. The MRMR algorithm uses an additional forward process to determine the feature rankings, which requires O(|ϕ|·|S|) computation of the mutual information index (MIQ) values.(7)$${MIQ}_{x}=\frac{MA_{x}}{MI_{x}}$$ Here, $MA_{x}$ and $MI_{x}$ are the relevance and redundancy of a feature, respectively, as(8)$$MA_{x}=I\left(x,y\right),\;MI_{x}=\frac{1}{\left|S\right|}\sum_{z\in S}I\left(x,z\right)$$ The MRMR algorithm assigns a weight order to all features of ϕ and returns a feature index sorted by feature weight. Here, the computational cost is $O(|\phi {|}^{2})$. In addition, the MRMR determines the importance and weight of features empirically. Therefore, a vital weight value means the feature validates predicting the response variable. Furthermore, decreasing the weight value of the features increases the confidence in the feature determination. Thus, the output can be used to find a valid feature set *S* for a given number of features, as shown in the Algorithm 1.(9)$$\max_{x\in S^{c}}{MIQ}_{x}=\max_{x\in S^{c}}\frac{I(x,y)}{\sum_{z\in S}I\left(x,z\right)}$$
**Algorithm 1** MRMR(X, Y)1:$(\max_{x\in \phi }MA_{x}$): determine the most relevant features;2:(x⊂S): including selected feature *x* in empty set *S*;3:**do**4:   ($MA_{x}\neq 0,\in S^{c},MI_{x}=0,\in S^{c}$): finding, where $S^{c}$ is the complement of *S*;5:   **if** ($MA_{x}\neq 0,\notin S^{c}$) and ($MI_{x}=0,\notin S^{c}$) go to line 12:6:   **else**7:      ($\max_{x\in S^{c},MI_{x}=0}MA_{x}$): determine most relevance feature;8:      (x⊂S): including selected feature *x* to the set *S*;9:   **end**10:**while** until $MI_{x}\neq 0$ for ∀ feature $\in S^{c}$;11:**do**12:   ($\max_{x\in S^{C}}MIQ_{x}$): determine feature with the nonzero relevance and redundancy;13:   (x⊂S): including the selected feature *x* to the set *S*;14:**while** until $MA_{x}=0$ for ∀ feature $\in S^{c}$;15:   ($MA_{x}=0$): including feature to the set *S* in random order (x⊂S);

### 3.5. GBA

Given GBA is a powerful constraint for researchers because appropriate target values must be provided for the data [33]. Here, we combine MRMR with the GBA model to determine the feature set for the best BP estimation results. The algorithm WFD (Algorithm 2) process is to select the weighted feature subset with the smallest root mean square error using GBA, as shown in the algorithm. Therefore, input data and target data need to be prepared as ${\left(x_{i},y_{i}\right)}_{i=1}^{N}$. Our goal is to minimize the expectation of a given cost function Ω(y,F(x)), given by the following equation, and obtain an estimate $F^{*}\left(x\right)\approx y$ as(10)$$F^{*}\left(x\right)=\arg\min_{F(x)}\Omega \left(y,F\left(x\right)\right)$$
where $F^{*}\left(x\right)$ denotes a boosting approximate which is calculated by $F\left(x\right)=\sum_{m=0}^{M}\alpha_{m}h\left(x,a_{m}\right)$, here $h(x,a_{m})$ denotes a base learner, $\alpha_{m}$ are the coefficients, and $a_{m}$ denote the parameters. Target data are a continuous variable, i.e., y∈R, we use the classical cost function.(11)$$\left(\alpha_{m},a_{m}\right)=\arg\min_{\alpha ,a}\sum_{i=1}^{N}\Omega \left(y_{i},F_{m-1}\left(x_{i}\right)\right)+\alpha h\left(x_{i},a\right)$$(12)$$F_{m}\left(x\right)=F_{m-1}\left(x\right)+\alpha_{m}h\left(x,a_{m}\right)$$ Here, the step size α is specified at each iteration, and $a_{m}$ represents the *m*th incremental step of the parameter. Generally, it is challenging to obtain parameter estimates when only the cost function Ω(y,f(x)) and the base learner $h(x,a_{m})$ are given. Therefore, we can use the negative gradient ${\left\{g_{m}\left(x_{i}\right)\right\}}_{i=1}^{N}$ as(13)$$-g_{m}\left(x_{i}\right)=-{\left[\frac{\partial \Omega (y_{i},F\left(x_{i}\right))}{\partial F(x_{i})}\right]}_{F\left(x\right)=F_{m-1}\left(x\right)}$$ Here, h(x,a) is the most correlated with $-g_{m}$ in terms of data distribution. Hence, we can solve as(14)$$a_{m}=\arg\min_{\alpha ,a}\sum_{i=1}^{N}{\left[-g_{m}\left(x_{i}\right)-\alpha h\left(x_{i},a\right)\right]}^{2}$$ Here, the constrained negative gradient $h(x_{i},a)$ is utilized in place of the $-g_{m}\left(x_{i}\right)$ in the steepest-descent strategy. Thus, (Equation 11) can be defined as (Equation 15).(15)$$\beta_{m}=\arg\min_{\beta }\sum_{i=1}^{N}\Omega \left(y_{i},F_{m-1}\left(x_{i}\right)+\beta h\left(x_{i},a_{m}\right)\right)$$ The estimated is updated as(16)$$F_{m}\left(x\right)=F_{m-1}\left(x\right)+\beta_{m}h\left(x,a_{m}\right)$$
**Algorithm 2** Weighted feature decision (WFD)1:**call** MRMR2:[w-score, idx] = MRMR(X, Y): dataset.3:bestmse = zeros(*n*, 2);4:**for** i = 1,n; where n denotes the number of feature dimension5:   m = 5; where m denotes the number of folds for cross-validation.6:   rmse = zeros(1,m);7:   cv = cv-partition(length(y),‘kfold’,m); where cv is cross-validation.8:   num-iter = 50;9:   **for** k = 1,m10:      xtrain = X(cv.train(k), :);11:      ytrain = Y(cv.train(k), :);12:      xtest = X(cv.test(k), :);13:      ytest = Y(cv.test(k), :);14:      **call** GBA(xtrain(:, idx(1:n)), Ytrain, num-iter, parameters);15:      return(mdp); where mdp is the return parameter of GBA model16:      predict(‘mdp’, xtest(:, idx(1:n)));17:      return(ys);18:      mse(k) = $\left(\mathrm{mean}\left({\left(\mathrm{ys}-\mathrm{ytest}\right)}^{2}\right)\right)$;19:   **end for**20:   β = mean(mse);21:   bestmse(i, 1) = β22:   bestmse(i, 2) = n;23:   n = n − 1;24:**end for**25:[index] = min(bestmse(:, 1));26:n = 44;27:n = n-index + 1;28:decision(idx(1:n));

A regression tree is a variant of a decision tree that predicts the error. Therefore, the regression tree model attempts to find a relationship between the dependent and independent variables separated from the initial dataset. The input features are given as (X,Y), where X and *Y* represent the independent and dependent variables, respectively. GBA aims to obtain a multi-estimator given $\left\{F_{1}\left(x\right),\ldots ,F_{M}\left(x\right)\right\}$ to overcome unstable estimates such as high variance and error bias [33]. Here, we first show how to build a regression tree to optimize GBA and then show the process of learning relevant features while growing the tree.

It should be noted that the parameter space of GBA is less practical than grid search. Therefore, we use a closed-form kernel that can handle large-sized parameters. In general, GBA finds a set of base learners that depend on a fixed set of features and selects $h_{j}$ that is minimized at each iteration *j*.(17)$$h_{j}=\arg\min_{h}\sum_{i=1}^{N}w_{i}^{j}{\left[h\left(x_{i}\right)-\theta_{i}^{j}\right]}^{2}$$ Here, $w_{i}^{j}$ are the weights, and $\theta_{i}^{j}$ is the dependent variable computed by differentiating the cost function Ω. The baseline learner is a regression tree that uses the convolution of the learned convolutional kernel set $\Phi_{j}$ and x. Therefore, $h_{j}\left(x\right)$ can be redefined as $\Psi (x,\Phi_{j},\delta_{j})$, where $\delta_{j}$ is the tree parameter, and the tree learning step is performed one split at a time. In addition, the split consists of a validation function v(·)∈R, a threshold τ, and return parameters $\zeta_{1}$ and $\zeta_{2}$. Therefore, the estimation function can be expressed as follows:(18)$$p\left(\cdot \right)=\left\{\begin{array}{ccc} \zeta_{1} & \mathrm{if} & v(\cdot )<\tau \\ \zeta_{2} & \; & \mathrm{otherwise} \end{array}\right.$$ We can detect the optimal split by minimizing v(·) at iteration *j* given(19)$$\sum_{i|v(x_{i})<\tau }w_{i}^{j}{\left(\theta_{i}^{j}-\zeta_{1}\right)}^{2}+\sum_{i|v(x_{i})>\tau }w_{i}^{j}{\left(\theta_{i}^{j}-\zeta_{2}\right)}^{2}$$ Here, τ, $\zeta_{1}$, and $\zeta_{2}$ are found using a grid search. The validation function $v\left(x_{i}\right)=\phi^{T}x_{i}$ is used, which is run on the output of $x_{i}$ and the kernel $\phi_{l}$. In addition, segmentation learning requires finding the kernel ϕ, the leaf parameters $\zeta_{1}$ and $\zeta_{2}$, and the segmentation threshold τ that minimizes $v(x_{i})$ (Equation 19). First, we build a set of kernel candidates. Then for each candidate, we can use grid search to find the optimal threshold τ, and for a given kernel ϕ and threshold τ, the optimal values for $\zeta_{1}$ and $\zeta_{2}$ can be found as the weighted average of the values of the $x_{i}$ samples $\theta_{i}^{j}$ on the corresponding side of the split.

To facilitate the task, we restrict the kernel ϕ to a square window within x, which is a more general treatment than most previous methods. This method reduces the problem’s dimensionality and allows the department to focus on local features. An operator $W_{c,a}\left(x_{i}\right)$ that returned the pixel values of x in vector form within a square window centered at *c* on side length *a*, was introduced. The criteria in (Equation 17) were as follows:(20)$$\sum_{i=1}^{N}w_{i}^{j}{\left(\phi^{T}W_{c,a}\left(x_{i}\right)-\theta_{i}^{j}\right)}^{2}$$ Here, ϕ is restricted to a square window parameterized by *c* and *a*. Given *c* and *a*, the optimal ϕ can be computed in closed form by solving the least-squares problem in (Equation 20). Two improvements are introduced to prevent overfitting: regularization and training set splitting. The term regularization is used to favor smooth kernels in the criterion given in (Equation 20)(21)$$\sum_{i}w_{i}^{j}{\left(\phi^{T}W_{c,a}\left(x_{i}\right)-\theta_{i}^{j}\right)}^{2}+\rho \sum_{(m,n)\in N}{\left(\phi^{m}-\phi^{n}\right)}^{2}$$ Here, (m,n)∈N denotes the index pairs corresponding to the neighboring pixels, and $\phi^{m}$ denotes the *m*th pixel of the kernel ϕ. The second term in Equation (Equation 21) imposes a smooth kernel controlled by ρ ≥ 0. Therefore, (Equation 21) can be minimized in closed form using the least-squares method.

## 4. GPR

GPR is a powerful and flexible non-parametric method that can be utilized for supervised ML. This is beneficial when there is underlying Gaussian distribution [15]. First, we examine the Bayesian analysis of linear regression, which projects the input into a high-dimensional feature space and applies linear regression. The training input and output datasets are $D={\left\{x_{i},y_{i}\right\}}_{i=1}^{I}$, $x\in R^{I\times D}$, and $y\in R^{I\times 1}$. Then, we estimate the dependent variable *y* from the given independent variable *x* by applying the mapping function $f_{m}=f\left(x\right)$. Therefore, we suppose that the dependent variable *y* can be obtained using the given $x^{T}w$, including noise, as follows(22)$$y=x^{T}w+\varepsilon ,\;\varepsilon ∽N\left(0,\sigma^{2}I\right)$$ In the above (Equation 22), the variance $\sigma^{2}$ and weight vector *w* are obtained from the dataset. The GPR can predict the dependent variable *y* based on the Gaussian process (GP) using the mapping function $f_{m}\left(x\right)$ and the explicit essential function β.(23)$$f_{m}\left(x\right)∽\mathrm{GP}\left(0,k\left(x,x^{\prime }\right)\right)$$
where $f_{m}\left(x\right)$ is obtained from a zero-mean GPR algorithm based on a covariance function $k(x,x^{\prime })$ [34]. Thus, we acquire the mapping function given as $f_{m}\left(x\right)=\beta {\left(x\right)}^{T}w$. The mean value of the input data is represented by the expectation of the mapping function $\theta \left(x\right)=E[f_{m}\left(x\right)]$, where the latent variable covariance function obtains the smoothness of the dependent variable, and the basis function requires the projection of the input data *x* into a dimensional feature space.(24)$$k\left(x,x^{\prime }\right)=E\left[\left(f_{m}\left(x\right)-\theta \left(x\right)\right){\left(f_{m}\left(x^{\prime }\right)-\theta \left(x^{\prime }\right)\right)}^{T}\right]$$ Thus, the expected value of (Equation 24) as(25)$$k\left(x,x^{\prime }|\eta \right)\approx \sigma^{2}\exp\left(-\frac{{∥x-x^{\prime }∥}^{2}}{2\eta^{2}}\right)$$
where k shows a kernel for the GPR [15], η denotes a hyperparameter, and $\sigma^{2}$ shows a variance using input signals. Here, the kernel function uses exponential squares, as in (Equation 25). Therefore, we can use the kernel to determine the properties of the mapping function $f_{m}\left(x\right)$. In addition, we use the Bayesian inference-based GPR to define the instances of the dependent variable y as follows(26)$$p\left(y_{i}|f_{m}\left(x_{i}\right),x_{i}\right)∽N\left(y_{i}|\beta {\left(x_{i}\right)}^{T}w+f_{m}\left(x_{i}\right),\sigma^{2}\right)$$ Here, $\beta (x_{i})$ is the basis function that transforms the original dependent variable *x* into the new dependent variable β(x). Therefore, from the dataset D containing the input values and dependent variables, we determine $\Theta =\{w,\eta ,\sigma^{2}\}$, and the boundary likelihood can be expressed as(27)$$p\left(y|x\right)=p\left(y|x,\Theta \right)\approx N(y|\Omega w,k\left(x,x^{\prime }|\eta \right)+\sigma^{2}I),$$ We can train the GPR algorithm by determining the local maximum for the hyperparameter Θ. Also, we may predict the expected patterns and smoothness of the data by selecting an appropriate kernel. Based on maximizing the log marginal likelihood, we predict the hyperparameter Θ, as(28)$$\begin{array}{c} \log p\left(y|x,\Theta \right)=-\frac{1}{2}\log\left|k\left(x,x^{\prime }|\eta \right)+\sigma^{2}I\right|-\frac{1}{2}i\log2\pi \\ \;-\frac{1}{2}{\left(y-\Omega w\right)}^{T}{\left[k\left(x,x^{\prime }|\eta \right)+\sigma^{2}I\right]}^{-1}\left(y-\Omega w\right) \end{array}$$ Here, $k(x,x^{\prime }|\eta )$ presents the kernel matrix and Ω represents the matrix of explicit basis functions. Here, we express the algebraic likelihood by applying the penalty fitting scale and maximizing it utilizing the gradient approach with optimization techniques. The hyperparameter $\Theta =\{w,\eta ,\sigma^{2}\}$ based on the GPR algorithm maximizes the likelihood p(y|x) as a function of Θ.(29)$$L\left(\hat{\Theta }\right)=\arg\max_{\Theta }\log\left(y|x,\Theta \right)$$ First, we determine $\hat{w}\left(\eta ,\sigma^{2}\right)$ to predict hyperparameters that maximize the log-likelihood concerning *w* for a given $\left(\eta ,\sigma^{2}\right)$ as(30)$$\hat{w}\left(\eta ,\sigma^{2}\right)={\left\{\Omega^{T}{\left[k\left(x,x^{\prime }|\eta \right)+\sigma^{2}I\right]}^{-1}\Omega \right\}}^{-1}\Omega^{T}{\left[k\left(x,x^{\prime }|\eta \right)+\sigma^{2}I\right]}^{-1}y$$ Second, we use the probability density function $p(y^{*}|y,x,x^{*})$ for the probabilistic estimation of the Bayesian GPR algorithm using hyperparameters. However, we can estimate the dependent variable *y* using a finite amount of new independent variables $x^{*}$ and predict the output of these data based on a multivariate Gaussian distribution with a kernel-generated covariance matrix. Therefore, the conditional probability distribution is expressed as follows(31)$$p\left(y^{*}|y,x,x^{*}\right)=\frac{p(y^{*},y|x,x^{*})}{p(y|x,x^{*})}$$ To obtain the numerator’s probability density function as shown in (Equation 31), we need to use the mapping functions $f_{m}^{*}$ and $f_{m}$ as follows(32)$$\begin{array}{c} p\left(y^{*},y|x,x^{*}\right)=\int \int p\left(y^{*},y,f_{m}^{*},f_{m}|x,x^{*}\right)dfdf^{*}\\ \;=\int \int p\left(y^{*},y|f_{m}^{*},f_{m},x,x^{*}\right)p\left(f_{m}^{*},f_{m}|x,x^{*}\right)dfdf^{*} \end{array}$$ The GPR algorithm assumes that each response variable $y_{i}$ depends only on the corresponding latent variable $f_{m}\left(x_{i}\right)$ and input vector $x_{i}$ (see [35] for a detailed derivation of (Equation 32)). Given y,x and the hyperparameters Θ, the expected value of the estimation is given as:(33)$$\begin{array}{cc} E(y^{*}|y,x,x^{*},\Theta ) & =\theta {\left(x^{*}\right)}^{T}w+c\left(x,x^{\prime }|\eta \right)\varphi \\ \; & =\beta {\left(x^{*}\right)}^{T}w+\sum_{i=1}^{I}\varphi_{i}k\left(x^{*},x_{i}|\eta \right) \end{array}$$
where $\varphi ={\left[k\left(x,x\right)+\sigma^{2}I\right]}^{-1}\left(y-\Omega w\right)$, Practically, we determined an optimal point prediction ${\hat{y}}^{*}$ based on the loss function, as(34)$$E_{L}\left({\hat{y}}^{*}|x^{*}\right)=\int L\left(y^{*},{\hat{y}}^{*}\right)p\left(y^{*}|x^{*},D\right)dy^{*}$$ We obtained and prediction $y^{*}\approx {\hat{y}}^{*}$ and minimized the expected value of the loss function $L(y^{*},{\hat{y}}^{*})$ by minimizing between $y^{*}$ and ${\hat{y}}^{*}$ as(35)$${\hat{y}}_{\mathrm{opt}}|x^{*}=\arg\min_{{\hat{y}}^{*}}E_{L}\left({\hat{y}}^{*}|x^{*}\right)$$ This study uses the mean absolute error (MAE) as the loss function, which is given by L.

## 5. Uncertainty Estimation for Individual Subject

### 5.1. Measurement Uncertainty

The purpose of measurement is to obtain the actual value of the measurand. The quantity to be measured is called the measurand [4]. We cannot know precisely how close the measured BP value is to the actual BP value. Therefore, BP estimates always contain uncertainty. The difference between the estimated and actual BP values is called the error, a well-known source of uncertainty. Here, uncertainty quantifies the doubt about the BP measurement result [20]. Considering the BP error, which consists of two parts, systematic error and random error, we cannot know the BP error precisely because we do not know the actual BP value. Therefore, evaluating the BP measurement result without considering the uncertainty would be challenging. The quality and accuracy of the BP measurement result are characterized by its uncertainty, which defines the interval around the measured BP value within which we can judge with some probability that the true BP value exists.

In addition, the uncertainty *U* of BP measurement is half the width of the interval and must always be positive [4]. We can consider uncertainty in BP measurement as an estimate of the most considerable absolute difference between the measured and actual BP values. BP measurement can be affected by two types of errors: systematic error and random error. Here, the characteristic of systematic error is that it causes consistent deviations in the same direction and magnitude in all BP measurements. Increasing the number of repetitions does not reduce the influence of systematic error, such as bias. On the other hand, the characteristic of random error can cause differences between the results of repeated BP measurements. However, as the number of BP measurements increases, the influence of random error on the mean value decreases, so increasing the number of repetitions of BP measurements can reduce the influence of random error.

One of the most common ways to increase the reliability of a measurement is to take repeated BP measurements of the same quantity. Basic statistical calculations can be performed to increase the amount of information obtained from continuous BP measurements. We use the arithmetic mean to estimate the actual BP value. It also tells us how wide the range of BP measurements is since the results of BP measurements vary. The distribution of the results of continuous BP measurements provides the user with information about the uncertainty and standard deviation (SD) $\hat{\sigma }$ of the BP measurement. Here, the SD is the criterion for defining the standard uncertainty, which is denoted by *u*. This standard uncertainty *u* is calculated as $u=\hat{\sigma }/\sqrt{n}$. Generally, if an uncertainty estimate is obtained from the SD of the results of continuous BP measurements, it is defined as a Type A uncertainty estimate. Any uncertainty estimate obtained without repeated, continuous BP measurements is defined as a Type B uncertainty estimate using an assumed probability distribution, which can be made through experience or information [4]. Once again, in the absence of the actual BP value, the establishment of a reference value is crucial. This reference value, far from being a simple concept, is a set of quantitative measurements that can be accurately expressed, therefore enhancing the accuracy of our measurements. One pivotal factor in determining the authenticity of the BP value is bias. This is not just a difference between the BP value measured repeatedly with the same sample and the reference value but a quantitative expression of authenticity.

In the context of BP measurement, we can use two properties to estimate uncertainty, which provides a quantitative measure of its accuracy. The combined measurement uncertainty for BP can be expressed in terms of bias, SD, and the combined standard uncertainty ($u_{c}$), as per the approach by Stergiou et al., which is given by the formula $u_{c}=\sqrt{u_{\alpha }^{2}+u_{\beta }^{2}+u_{\gamma }^{2}}$, where $u_{\alpha }$ represents the standard uncertainty as random error, $u_{\beta }$ represents the bias as systematic error, and $u_{\gamma }$ represents the maximum allowable error (which is the systematic error for a mercury sphygmomanometer, approximately ± one mmHg according to [36]), which systematic error is a significant source of uncertainty. Additionally, we provide CI to express and evaluate uncertainty. A CI is a kind of interval estimate calculated from repeated BP measurement statistics, which can include the actual BP value of an unknown population parameter. Therefore, the CI is used as an extended uncertainty, i.e., $U=K\times u_{c}$, and we can obtain the CI of the measurement object as $\bar{x}\pm U$ [20]. Here, we define $\bar{x}$ as the mean of the measurement object, and if the BP measurements’ distribution shows a Gaussian distribution, the arithmetic mean value is given as $\bar{x}$, and the standard uncertainty can be given as the SD of this distribution σ. If K=2, the CI is $\bar{x}\pm 2\sigma$, and the confidence level increases to 95%. Hence, we obtain ten segmented PPG signals from 20 (s) for each subject in the proposed algorithm. We estimate BP using individual features obtained from 10 segmented PPGs and generate uncertainty and CI for individual subjects using the mean BP and SD.

### 5.2. CI Estimation Using Bootstrap for Individual Subject

The bootstrap method is the cornerstone of our approach, using the uncertainty range of the continuous BP measurement value to calculate the maximum and minimum values of the CI. This is achieved by applying the bootstrap principle within the parametric approach, a connection that is crucial to our work. In our proposed algorithm, we acquire ten segmented PPG signals from 20 (s) for each subject in the proposed algorithm. We estimate BP using individual features obtained from 10 segmented PPGs and generate uncertainty, ${\hat{T}}^{*}=\left({\hat{t}}_{1}^{*},\ldots ,{\hat{t}}_{n}^{*}\right)$, based on *n* estimates obtained from an unknown distribution D(μ,σ) to compute a CI for $\hat{\mu }\left(T^{*}\right)$. Here, $\left[\hat{\mu },\hat{\sigma }\right]$ denotes the maximum likelihood estimate obtained using $\hat{T}=\left({\hat{t}}_{1},\ldots ,{\hat{t}}_{n}\right)$. Thus, when n→∞, we obtain a normal distribution given as $\hat{D}\left({\hat{\mu }}^{*},{\hat{\sigma }}^{*}|{\hat{T}}^{*}\right)≅N\left(\mu ,\sigma \right)$. In our work, we measure the CIs utilizing the bootstrap technique [6], Ref. [19] which can be obtained using the BP estimates of the proposed methodology. We then obtain a matrix as follows:(36)$$\begin{array}{c} M^{*}\left(i|{\hat{T}}_{i}^{S*}\right)=\left[\begin{array}{ccc} t_{1,1}^{*i} & \cdots & t_{1,B}^{*i}\\ \vdots & \ddots & \vdots \\ t_{n,1}^{*i} & \cdots & t_{n,B}^{*i} \end{array}\right] \end{array}$$
where (Equation 36) is acquired as ${\hat{\mu }}_{i}^{*}+{\hat{\sigma }}_{i}^{*}\times \mathrm{RANDN}(n,B)$, we then vertically compute each column to obtain the average of each column as ${\hat{\mu }}_{b}^{s*}=1/n\sum_{j=1}^{n}t_{j,b}^{*i}$, where s denotes SBP, and ∗ indicates the resampled data obtained from the bootstrap technique. We hence do ascending sorts, and the sorted BP estimate is given by ${\hat{\Xi }}^{s*}=\left({\hat{\mu }}_{1}^{s*},{\hat{\mu }}_{2}^{s*},\cdot \cdot \cdot ,{\hat{\mu }}_{B}^{s*}\right)$, assuming ${\hat{\mu }}_{\alpha }^{s*}$ is the 100α th percentile of *B* bootstrap replications $\left({\hat{\mu }}_{1}^{s*},{\hat{\mu }}_{2}^{s*},\cdot \cdot \cdot ,{\hat{\mu }}_{B}^{s*}\right)$. We acquire the CI as ${\hat{\mu }}_{\mathrm{lower}}^{s*},{\hat{\mu }}_{\mathrm{upper}}^{s*}$ of the 1−2·α, from this bootstrap technique, as $\left({\hat{\mu }}_{\alpha }^{s*},{\hat{\mu }}_{1-\alpha }^{s*}\right)$. Similar process is used to estimate the CI for DBP.

### 5.3. CL Estimation Using Monte Carlo for Individual Subject

The Monte Carlo method is to estimate a population expectation from corresponding sample expectation [37]. Hence, we can calculate CI for a sample arithmetic mean. Here, we suppose the expected value of a random variable given as $E({\hat{\mu }}_{i}^{S},{\hat{\sigma }}_{i}^{S}|{\hat{T}}_{i}^{S})$ in a Monte Carlo technique. Hence, we create values $\left({\hat{T}}_{1},{\hat{T}}_{2},\ldots ,{\hat{T}}_{n}\right)$ to be independent and identically distributed (IID) random variables from the distribution of $\hat{T}$ using the BP target results based on the proposed methodology and obtained their arithmetic mean and SD [4]. The SD of ${\hat{\mu }}_{i}^{s}$ is given as $\sqrt{E\left[{\left({\hat{\mu }}_{i}^{s}-\mu_{i}^{s}\right)}^{2}\right]}=\sigma /n$, which is the root mean square error of ${\hat{\mu }}_{i}^{s}$. However, we do not exactly know σ. Thus, we commonly use estimates of σ given by $\hat{\sigma }=\sqrt{\frac{1}{n-1}\sum_{i=1}^{n}{\left({\hat{T}}_{i}-{\hat{\mu }}_{i}^{s}\right)}^{2}}$. Based on the central limit theorem given as $P\left(\lim_{n\to \infty }|{\hat{\mu }}_{i}-\mu |=0\right)=1.$ Here, we assume μ as the reference value measured by the two nurses. Therefore, the probability density function is calculated by $P\left(z\right)=\frac{1}{\sqrt{2\pi }}\exp^{-\frac{1}{2}z^{2}}$ and a cumulative distribution function computed by $\Psi \left(a\right)=\int_{-\infty }^{a}P\left(z\right)dz$.

**Theorem** **1.**
*If $\left\{{\hat{T}}_{i}\right\}$ is IID with mean μ and variance $\sigma^{2}>0$ [4], then for all z∈R*

(37)
$$P\left(\sqrt{n}\frac{{\hat{\mu }}_{i}-\mu }{\sigma }⩽z\right)⟶\Psi \left(z\right)$$

*We compute the CI through the theorem for μ; however, it requires that we know σ. P(∣s−σ∣ >ε) becomes 0 as n⟶∞. As a result, we can use s for σ. We restate Equation (Equation 37) as*

(38)
$$\begin{array}{cccc} P\left(|{\hat{\mu }}_{i}-\mu |⩾\frac{\varepsilon s}{\sqrt{n}}\right) & =P\left(\sqrt{n}\frac{{\hat{\mu }}_{i}-\mu }{s}⩽-\varepsilon \right)+P\left(\sqrt{n}\frac{{\hat{\mu }}_{i}-\mu }{s}⩾\varepsilon \right) & \; & \\ \; & ⟶\Psi (-\epsilon )+(1-\Psi (-\epsilon )) & \; & \end{array}$$


(39)
$$\begin{array}{ccc} \; & =2\Psi (-\varepsilon )=0.05 & \; \end{array}$$

*Here ε>0. Therefore, we compute a 95% CI with $\varepsilon =-\Psi^{-1}\left(0.025\right)=\Psi^{-1}\left(0.975\right)$ as given by*

(40)
$${\hat{\mu }}_{i}-\Psi^{-1}\left(0.975\right)⩽\frac{s}{\sqrt{n}}⩽\mu ⩽{\hat{\mu }}_{i}+\Psi^{-1}\left(0.975\right)⩽\frac{s}{\sqrt{n}}.$$

*Then, we summarize the central limit theorem based-approximate CIs of the form ${\hat{\mu }}_{i}\pm \Psi^{-1}$$\left(1-\alpha /2\right)s/\sqrt{n}$ for 95 and 100(1−α)% CIs, respectively.*


## 6. Experimental Results

The parameters utilized in the proposed and conventional models were tuned using the validation stage. As mentioned in the introduction, learning models such as SVM [38], NN [10], regression tree (RTree) [39,40], GPR [15], and DNN [9,18] were used to compare the performance with the proposed combining DPFE and WFD based on GPR, which have been popular in cuff-less BP estimation until recently. The popular SVM models [38] were suitable for handling nonlinear relationships between features extracted from ECG and PPG signals and reference BP, as they can handle nonlinear relationships in input and output data well. Many researchers have used the NN models to estimate BP using PPG signals [10,41]. The random forest (RF) regression model used trees and integrated various for BP estimation [10,42]. The ensemble bagging was recently proposed for the PPG signal-based BP classification by Turnip et al. [43]. This year, the ensemble model was utilized for heart disease prediction using adaptive boosting, random forest, and extreme gradient boosting [44]. Different DNN models have recently been used for BP estimation based on PPG with ECG signals [8,9,18].

The tuning stage of parameters was crucial because it can considerably improve the performance of ML models. In this study, ten-fold cross-validation fine-tuned the parameters for all ML models. The main parameters for each model were defined, and the possible range of values was determined for each parameter, highlighting our research’s potential benefits and advancements. The process began with all ML models, including DNN [9,18], and RF [42] models, conducting a grid search for all possible parameter combinations to identify the parameters for achieving the best performance results. This ten-fold cross-validation was performed to enhance the robustness of the two models using the optimal parameters identified. The training data were then divided into ten non-overlapping subsets of equal sizes for iterative learning. Each iteration involved nine-fold training and assessment of the model using the remaining folds. Table 2 lists the optimized parameters for each ML model respectively.

The WFD process was developed by combining the GPR with GBA. The PPG signals were divided into 80% for training, 10% for verification, and 10% for testing. The reference BP was calculated from the ABP signal. The running times were estimated based on the entire dataset using MATLAB ^®^2024 [46] as shown in Table 3. The SVR [10], NN [47], DNN [9], regression tree (RTree) [40], and random forest (RF) [42] models served as benchmark algorithms to compare the performance of the proposed WFD based on the combined GPR and GBA. The mean error (ME), MAE, and standard deviation (SD) results presented the mean values of 30 experiments. Table 4 lists statistical information on the range and distribution of the reference SBP and DBP in the final dataset (59,097 records).

We compare the proposed WFD based on the combined GPR and GBA and the conventional methods using the SD of the ME to evaluate the experimental results as shown in Table 5. The ME and SD were determined based on the association’s recommendations for advancing medical instrumentation (AAMI) protocol [36].

The AAMI protocol recommends that the measurement is exceeded when the ME value is ≤5 mmHg and the SD is ≤8 mmHg. The European Society of Hypertension (ESH) recommendations have validated cuff-less BP measuring devices as the ME value is ≤5 mmHg and the SD is ≤8 mmHg [48]. The results, represented by the MAE, provide a reliable evaluation of the proposed algorithm, as shown in Table 6. We compare the MAEs of our proposed method and state-of-the-art models, as shown in Table 7.

In addition, the recommended probabilities for the British Hypertension Society (BHS) protocol [49] are calculated based on the MAE and SD results as represented in Table 8. The MEs were calculated ($ME=\frac{1}{n}\sum_{i=1}^{n}me_{i}$) as $me_{i}=\left(ep_{i}-rp_{i}\right)$ for each record *i*, where ep represents the estimated BP (SBP or DBP), and rp represents the reference BP. The MAEs were computed as ($MAE=\frac{1}{n}\sum_{i=1}^{n}\left|me_{i}\right|$). The SDs were calculated as ($\sqrt{\frac{1}{n-1}\sum_{i=1}^{n}{\left(ME-me_{i}\right)}^{2}}$).

**Table 6 bioengineering-12-00131-t006:** MAE and SD relative to the reference ABP for the conventional and proposed methods.

Method Combined (mmHg)	SVM [38]		NN [10]		DNN [45]		RTree [39]		RF [42]		GPR [15]		GPR [15]	
	SBP	DBP	SBP	DBP	SBP	DBP	SBP	DBP	SBP	DBP	SBP	DBP	SBP	DBP
	DPFE		DPFE		DPFE		DPFE		DPFE		DPFE		DPFE&WFD	
MAE	1.87	0.94	8.75	4.50	6.09	2.64	2.92	1.47	3.14	1.49	1.54	0.73	1.46	0.69
SD	0.08	0.02	0.23	0.06	0.21	0.08	0.00	0.00	0.02	0.01	0.01	0.01	0.03	0.01

**Table 7 bioengineering-12-00131-t007:** Comparison with the state-of-the-art models introduced in the previous studies.

Method	Dataset	SBP (mmHg)		DBP (mmHg)		Published
		MAE	SD	MAE	SD	
SVM [17]	MIMIC-II (910 samples)	8.54	10.90	4.34	5.80	2017
NN [16]	MIMIC-II (2000 samples)	4.24	5.05	4.81	6.37	2021
RF [50]	Private DB (90 samples)	7.36	7.15	5.43	5.30	2024
DNN (MLPLSTM) [51]	MIMIC-II (3000 samples)	4.39	6.43	2.54	3.76	2022
DNN (RES-FRN) [52]	MIMIC III (833 subjects)	5.59	5.78	3.35	3.80	2023
DNN (CNN-LSTM) [53]	MIMIC III (431 subjects)	4.15	5.83	2.33	3.16	2024
DNN (CNN-Transformer) [54]	Aurora-BP DB (1125 subjects)	5.25	n/a	6.10	n/a	2024
DNN (Transformer+encoders) [8]	MIMIC III (1000 subjects)	3.91	5.65	2.29	3.01	2025
DNN (CNN+Transformer) [55]	MIMIC III (808 subjects)	4.44	5.98	2.36	3.22	2025
GPR DPFE & WFD	MIMIC-II (3000 subjects, 59,097 samples)	1.46	0.03	0.69	0.01	

We also offer the expanded uncertainty, and all were easily calculated based on the approaches detailed in GUM [20] using the bias and standard error for cuff-less BP estimation as shown in Table 9. Overall, the conventional ML models, including the GPR combining DPFE, show superior results than those without DPFE combination in all ML models, as shown in Table 5, Table 6 and Table 7.

## 7. Discussion

This paper is the first study to propose combining DPFE and WFD based on GPR for cuff-less uncertainties and BP estimations. Algorithm showed the high-weight features selected using MRMR with GBA models in the WFD method. Hence, the proposed methodology improves the cuff-less BP estimation compared to the conventional ML algorithms. Table 3 shows that the proposed DPFE and WFD using MRMR and GBA algorithms based on GPR require more computational resources than DPFE based on the GPR, indicating higher computational complexity. This also means that significant computational resources are used to accurately identify the optimal weighted feature combination to estimate cuff-less BP.

Based on the AAMI/ESH/ISO protocol and ESH evaluation results [36,48], the SVM represented the SD of ME for SBP (4.30 mmHg) and DBP (2.18 mmHg). The SD of ME for SBP (12.17 mmHg) and DBP (6.89 mmHg) obtained using the NN algorithm was compared with the reference BP. The SVM algorithm results are satisfactory and instilled confidence in its performance in evaluating the AAMI/ESH/ISO and ESH protocols. In contrast, the NN algorithm revealed that the SBP estimation results are unsatisfactory for the evaluation protocol as denoted in Table 5. The SDs of ME for SBP (7.92 mmHg and 5.76 mmHg) and DBP (3.68 mmHg and 2.96 mmHg) were obtained using DNN and RTree algorithms, respectively. The SBP and DBP results of the DNN algorithm barely passed the evaluation criteria, while the RTree algorithm also passed the evaluation criteria comfortably for the SBP and DBP results. Both algorithms passed the evaluation criteria, but the RTree algorithm perform better than the DNN algorithm, as shown in Table 5.

We represented the SD results of ME for SBP (5.08 mmHg) and DBP (2.96 mmHg) obtained using the RF algorithm [42], which had excellent learning ability. The RF satisfies the evaluation criteria. In addition, the SD of the ME was SBP (3.18 mmHg) and DBP (1.59 mmHg) in the GPR algorithm. The proposed combining DPFE and WFD based on GPR methodology is a significant leap, with a lower SD of ME for SBP (2.94 mmHg) and DBP (1.50 mmHg) compared to the existing SVM, NN, DNN, RTree, RF, and GPR algorithms as denoted in Table 5. The potential for cuff-less BP prediction to outperform conventional algorithms is the cause of excitement and optimism. These results indicate superior performance compared with conventional algorithms. Therefore, we conclude that this decreases the uncertainty, such as the SD of ME, and increases the performance reliability.

The proposed combining DPFE and WFD based on GPR methodology, differed minimally from the reference SBP and DBP. The agreement limits indicated by the red horizontal lines in Figure 4a,b are ME ± 2 × SD. Figure 4a shows the Bland–Altman plot of SBP, with the ME of −0.13 mmHg and the ±2.94 mmHg. Figure 4b presents the Bland–Altman plot of DBP, with the ME of −0.04 mmHg and the ±1.50 mmHg. These results demonstrate that the cuff-less BP estimated using the proposed combining DPFE and WFD based on GPR methodology passed to the AAMI/ESH/ISO and ESH protocols of ME ≤5 mmHg and the SD ≤ 8 mmHg. The SDs of the proposed method’s SBP and DBP were densely clustered within a narrow range compared with the reference ABP. The reliability of the RF algorithm is highlighted in Figure 4c,d, comparing its SD of ME for SBP and DBP with the reference ABP. The SD of the RF demonstrates consistent and reliable performance when compared to the reference BP (mmHg), assuring its performance.

Considering the evaluation results using the MAE, the SVM algorithm produced excellent SBP (1.87 mmHg) and DBP (0.94 mmHg) results compared to the reference BP presented in Table 6. The NN algorithm, with MAE of SBP (8.75 mmHg) and DBP (4.50 mmHg), and the DNN algorithm, with MAE of SBP (6.09 mmHg) and DBP (2.64 mmHg), are promising compared with the reference BP. However, the MAE results of the NN and DNN algorithms are relatively high compared to the proposed and conventional methods. The RTree algorithm demonstrates reliability with good MAE results for the SBP (2.92 mmHg) and DBP (1.47 mmHg) (Table 6). Similarly, MAE results of the RF algorithm were satisfactory SBP (3.14 mmHg) and DBP (1.49 mmHg) compared to the reference BP. These results confirm the reliability of these algorithms for BP evaluations. Finally, MAE results obtained using GPR were SBP (1.54 mmHg) and DBP (0.73 mmHg) compared to SBP (1.46 mmHg) and DBP (0.69 mmHg) obtained using the proposed WFD method which combined GPR and GBA. The proposed algorithm has the best results among the conventional algorithms mentioned above, confirming the effectiveness of WFD, which determines the weighted feature subsets. In addition, when the proposed WFD method was compared with the GPR algorithm, it showed a performance improvement of 5.5% = (1.54 − 1.46)/1.46 × 100 for SBP and 5.8% = (0.73 − 0.69)/0.69 × 100 for DBP. The SDs of the MAEs in all the algorithms are stable, as shown in Table 6.

We also denoted the conventional and state-of-the-art models on the MIMIC database to test the effectiveness of the proposed DPFE and WFD based on the GPR methodology. The comparison is shown in Table 7. The comparison included the database, the number of subjects, and each model’s final MAE and SD results. The DNN-based models [8,54,55] had more minor errors than the conventional ML models, but the MAE of SBP and DBP were higher than 3.91 mmHg and 2.29 mmHg, respectively, and the SD of SBP and DBP were higher than 5.05 mmHg and 3.01 mmHg, respectively. As a result, we confirm that the proposed combining DPFE and WFD based on GPR methodology shows superior results compared to the ML and DNN models in Table 7. When analyzing the MAE and SD in Table 6, the conventional methods using the DPFE also show better performance than the MAE and SD results of the conventional methods in Table 7. Hence, the proposed DPFE method contributes to the performance improvement of the conventional ML models, as shown in Table 6.

To test the effectiveness of the proposed combination of DPFE and WFD based on GPR methodology, we compare the literature results of existing and state-of-the-art models using the MIMIC database, which are presented in Table 7. The comparison included the database, the number of subjects, and each model’s final MAE and SD results. Although the DNN-based models [8,54,55] had slightly more errors than the existing ML models, the MAE of SBP and DBP were higher (3.91 mmHg and 2.29 mmHg), and the SD of SBP and DBP were higher (5.05 mmHg and 3.01 mmHg), respectively. As a result, we confirm that the proposed combining DPFE and WFD based on GPR methodology showed superior results compared to the ML and DNN models in Table 7. When we analyze the MAE and SD in Table 6, the conventional ML models using DPFE perform better than the MAE and SD results of the existing method in Table 7. Therefore, we confirm that the proposed DPFE method in Table 6 contributed to the performance improvement of the conventional ML models. We think the reason for the improved MAE performance is not only due to the influence of the DPFE and WFD combined model but also because the MIMIC data used in this study was much larger than that used in other literature. The big MIMIC data addresses the overfitting of the learning algorithm and ultimately contributes to performance improvement.

Additionally, we compared the proposed methodology with various algorithms, including the ensemble SVM, NN, DNN, RTree, RF, and GPR algorithms, following the recommendations of the BHS [49]. We obtained the MAE for three groups using each algorithm: <5 mmHg, <10 mmHg, and <15 mmHg. The results obtained using the proposed WFD methodology that combines GPR and GBA method were 94.18% (≤5 mmHg), 98.26% (≤5 mmHg), and 99.37% (≤15 mmHg) for SBP, and 98.27% (≤5 mmHg), 99.66% (≤10 mmHg), and 99.88% (≤15 mmHg) for DBP. The resulting probability of the proposed method based on the BHS evaluation protocol [49] is higher than that obtained using the existing SVM, NN, DNN, RTree, RF, and GPR algorithms as shown in Table 8. Compared with the BHS protocol [49], the proposed methodology received an A grade in SBP and DBP estimation, demonstrating a high probability and indicating its excellent performance. Furthermore, the proposed methodology is more accurate for cuff-less BP estimation than conventional algorithms. In addition, most algorithms, including the proposed algorithm, are superior with respect to the BHS evaluation criteria, as shown in Table 8.

Conventional cuff-less BP monitoring devices only provide estimates at a single point without a CI representing uncertainty. These devices imply their inability to distinguish statistical variations from variations due to physiological processes [56]. If this new method is used for cuff-less BP monitoring devices, wide CIs could trigger alarms, alerting the nurse station or primary care physician about potential patient risks in a home-based monitoring setting [56]. Hence, predicting the CI for cuff-less BP monitoring is crucial for improving reliability. This study is essential because it is the first to predict individual uncertainty using the proposed DPFE and WFD method, which combines GPR and GBA, showing that almost all BP estimates fall within the CIs based on test samples from all subjects in Figure 5. Thus, Figure 5 has substantial fluctuations. Table 9 shows that the proposed methodology can estimate uncertainty and potentially require a physician’s diagnosis of a health problem when the estimated BP exceeds the uncertainty range. The fourth row details the CI and BP estimates provided by GPR, which are 10.8 (mmHg) for SBP and 5.9 (mmHg) for DBP based on all subjects. The fifth row GPR values are obtained using the GPR with a bootstrap (${\mathrm{GPR}}_{\mathrm{Boot}}$) method based on all subjects. Here, the CI of SBP is 4.7 (mmHg), and the CI of DBP is 3.0 (mmHg). The 6th row GPR values are acquired using the GPR with an uncertainty (${\mathrm{GPR}}_{\mathrm{Uncer}}$) method based on all subjects. Here, the CI of SBP is 8.9 (mmHg) and the CI of DBP is 6.4 (mmHg). Hence, we can see that the CIs obtained using the (${\mathrm{GPR}}_{\mathrm{Uncer}}$) method are broader than that of (${\mathrm{GPR}}_{\mathrm{Boot}}$). Hence, the proposed ${\mathrm{GPR}}_{\mathrm{Boot}}$ and ${\mathrm{GPR}}_{\mathrm{Uncer}}$ methods provide narrower CIs than the method provided directly by GPR. Thus, they represent slightly more reliable uncertainties. However, since the CIs predicted using the Monte Carlo ${\mathrm{GPR}}_{\mathrm{Monte}}$ are too narrow, it is unlikely that the predicted BP value obtained using GPR, which is the expected BP value, will be included, so it is thought that there will be limitations in applying it to the system.

The values in the 7th row are the uncertainty for the one subject CI based on ten samples using the (${\mathrm{GPR}}_{\mathrm{Boot}}$) method. Using this method, the CI for SBP is 3.9 (mmHg), and the CI for DBP is 1.6 (mmHg). The 8th row, which represents the uncertainty (${\mathrm{GPR}}_{\mathrm{Uncer}}$), is the one subject CI based on ten samples. Here, the CI for SBP is 5.4 (mmHg) and the CI for DBP is 4.2 (mmHg). The values in the 9th row are the uncertainty for the other subject CI based on ten samples using the (${\mathrm{GPR}}_{\mathrm{Boot}}$). The CI for SBP is 4.8 (mmHg), and the CI for DBP is 2.7 (mmHg). The 10th row, which represents the uncertainty (${\mathrm{GPR}}_{\mathrm{Uncer}}$), is the other subject CI based on ten samples. Here, the CI for SBP is 8.6 (mmHg), and the CI for DBP is 6.4 (mmHg), as shown in Table 9. Also, rows 7 and 8 show an example of CIs for one subject using the ${\mathrm{GPR}}_{\mathrm{Boot}}$ and ${\mathrm{GPR}}_{\mathrm{Uncer}}$, and the detailed BP estimates and CIs are shown in Figure 6a,b. The red dots in Figure 6a are the reference SBP. The black dots are the SBP estimated combining DPFE and WFD based on GPR, where the blue dotted line is the CI upper value of the SBP obtained from the proposed ${\mathrm{GPR}}_{\mathrm{Boot}}$, and the green dotted line is the CI lower value of the SBP from the proposed ${\mathrm{GPR}}_{\mathrm{Boot}}$. Here, the magenta dotted line is the CI upper value of the SBP obtained from ${\mathrm{GPR}}_{\mathrm{Uncer}}$, and the cyan dotted line is the CI lower value of the SBP obtained from ${\mathrm{GPR}}_{\mathrm{Uncer}}$. Figure 6b shows the individual CIs of the DBP obtained by the proposed method. Figure 6a,b represent that almost all the individual BP estimates are included in the CIs of the proposed method.

The rows 9 and 10 present another example of CIs for another subject using the ${\mathrm{GPR}}_{\mathrm{Boot}}$ and ${\mathrm{GPR}}_{\mathrm{Uncer}}$, and the detailed BP estimates and CIs are shown in Figure 6c,d. The red dots in Figure 6c denote the reference SBP. The black dots denote the SBP estimated combining DPFE and WFD based on GPR, where the blue dotted line is the CI upper value of the SBP obtained from the proposed ${\mathrm{GPR}}_{\mathrm{Boot}}$, and the green dotted line is the CI lower value of the SBP from the proposed ${\mathrm{GPR}}_{\mathrm{Boot}}$. Here, the magenta dotted line shows the CI upper value of the SBP obtained from ${\mathrm{GPR}}_{\mathrm{Uncer}}$, and the cyan dotted line shows the CI lower value of the SBP obtained from ${\mathrm{GPR}}_{\mathrm{Uncer}}$. Figure 6d shows the individual CIs of the DBP obtained by the proposed method. Figure 6c,d represent that most of the individual BP estimates are included in the CIs of the proposed method. However, it should be noted that if the CI is too wide, the probability of including the BP estimate increases, but the reliability of the BP monitoring system decreases. Hence, the CIs of the BP estimation system are considered appropriate at 7–10 mmHg for SBP and 3–6 mmHg for DBP. Using the estimated CI, the proposed method can continuously monitor BP changes by assessing the uncertainty of cuff-less BP and hypertension risk. Therefore, combining DPFE and WFD based on GPR methodology is an effective health monitoring system, including cuff-less BP and CI estimation.

The overall results reveal that the proposed DPFE and WFD, combining GPR and GBA, are more accurate and highly reliable for cuff-less BP estimation. This reliability should instill confidence and reassurance among healthcare professionals, researchers, and developers in health monitoring systems. The proposed methodology can continuously monitor BP changes using the estimated CI to estimate the uncertainty of cuff-less BP and hypertension risk.

The proposed method providing lower SD, MAE, and CI suggests that the method performs better in accuracy and consistency. However, there are still situations where “false results” can occur, such as bias, overfitting, and narrow CI. If the proposed method has an inherent bias, even with low SD or MAE, it might consistently make errors in a specific direction (overestimating or underestimating), leading to false results. Also, suppose the proposed method is overfitted to the training data. In that case, it may produce accurate predictions on that specific data but fail to generalize well to unseen data, potentially leading to false results on new inputs. A narrow CI might suggest high confidence in predictions, but if the model is poorly specified or the assumptions are wrong, this could still lead to false or inaccurate results despite low SD and MAE.

### Limitations and Further Discussion

We performed experiments using the public MIMIC-II dataset (3000 subjects (59,097 records)) with some subject distribution limitations. In addition, the experimental results were required to satisfy the AAMI/ESH/ISO and ESH protocols [36,48]. However, the individual reference BP is linked to the reference BP based on the subject information provided by the MIMIC-II dataset. This study’s extensive MIMIC-II database records realistic physiological data of ten types of patients with noise or missing data gaps. Therefore, in future studies, we plan to use data from healthy people without BP disease when developing an algorithm for measuring BP in real life.

Another drawback is the complexity of combining DPFE and WFD based on GPR because it includes dual-step preprocessing as well as multiple feature extraction and selection processes for extracting valid features. Therefore, addressing this complexity is necessary in future studies. Our future research goal is to use the proposed algorithm in wearable devices. We perform cuff-less BP measurements locally on the patient’s wearable device. The cuff-less BP measurement device should be lightweight while providing accurate automated BP estimation and uncertainty measurements since wearable devices tend to have small and low-power processors. To improve the accuracy of the proposed algorithm, we will validate the proposed algorithm using a more significant number of data sets. We will also reduce the complexity of the proposed method so that it can be utilized in practical applications. We will discuss the proposed method’s complexity reduction in detail below.

The computational requirements of GPR can pose real-time challenges for wearable devices, especially when dealing with large data sets or high-dimensional problems. Here, we describe how to address the computational cost of GPR, which will be the focus of our future research. One powerful solution to these practical obstacles is sparse GPR. This approach allows us to approximate the posterior GP with a user-defined set of virtual training examples. This customization aspect will allow us to tailor the method to specific requirements by controlling the computational and memory complexity. The GPR field has provided a wealth of sparse approximations to overcome computational limitations. Many authors expect to accurately process only a subset of latent variables while providing approximate but computationally cheaper processing for the remaining variables [57].

A straightforward way to address the computational complexity of large data sets is to use a subset of data methods. This approach selects a smaller subset (m<n) of observations from the total n and then applies the GPR model to these m points for estimation while ignoring the other (n−m) points. This small subset is called the active set or the inductive input set. When dealing with many observations, using exact methods for parameter estimation and making predictions on new data can be expensive (exact GPR). One of the approximate methods to solve this is the block coordinate descent method [58], and we will apply this method in our study to analyze whether it is an efficient solution for the computational cost of GPR.

## 8. Conclusions

This study proposes a novel combining DPFE and WFD methods to estimate uncertainty and BP from PPG and ECG signals by combining GPR and GBA methodologies. The proposed method provides more accurate cuff-less BP and uncertainty estimation. The proposed DPFE and WFD method, which combines GPR and GBA, provides a superior model with improved reliability for cuff-less BP and uncertainty estimation by providing CI representing uncertainty. In future studies, we will conduct experiments using various datasets to increase the reliability of the proposed method for cuff-less BP and uncertainty. The proposed combining DPFE and WFD based on GPR methodology can improve accuracy and reliability in healthcare monitoring systems by estimating cuff-less BP and providing CI to represent uncertainty.

## Figures and Tables

**Figure 1 bioengineering-12-00131-f001:**
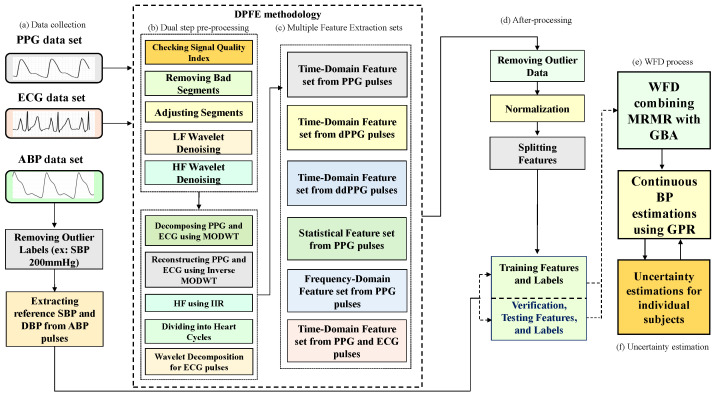
The lock diagram of proposed WFD based on GPR for cuff-less BP and CI estimations, where (**a**) is data collection, (**b**) denotes a dual-step preprocessing, (**c**) is multiple feature extraction sets, (**d**) presents after-processing, (**e**) denotes WFD process, and (**f**) presents uncertainty estimation using bootstrap and uncertainty algorithms for individual subject.

**Figure 2 bioengineering-12-00131-f002:**
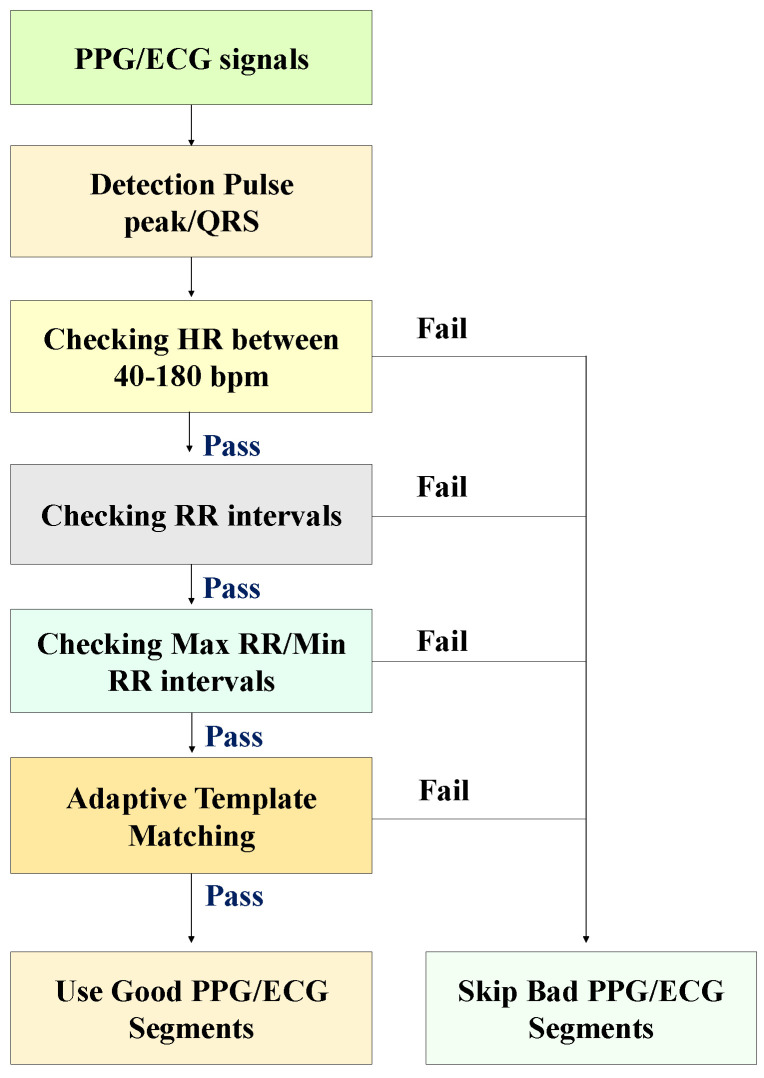
SQI algorithm for detecting bad PPG/ECG signals [30].

**Figure 3 bioengineering-12-00131-f003:**
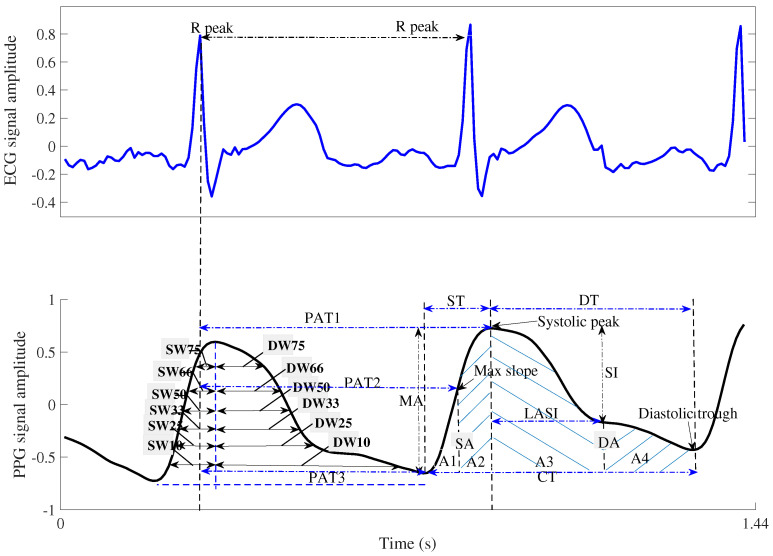
The features on each waveform of the PPGs and ECGs for cuff-less BP and CI estimations.

**Figure 4 bioengineering-12-00131-f004:**
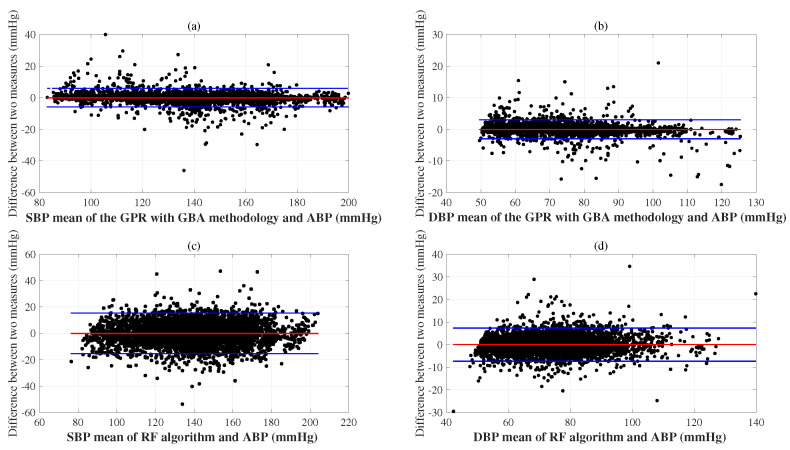
Bland–Altman plots for the proposed WFD combining GPR and GBA methodology represent very little difference from the referenced SBP (**a**) and DBP (**b**); the RF algorithm comparing its performance using reference ABP (mmHg) with respect to the SD of ME for SBP (**c**) and DBP (**d**).

**Figure 5 bioengineering-12-00131-f005:**
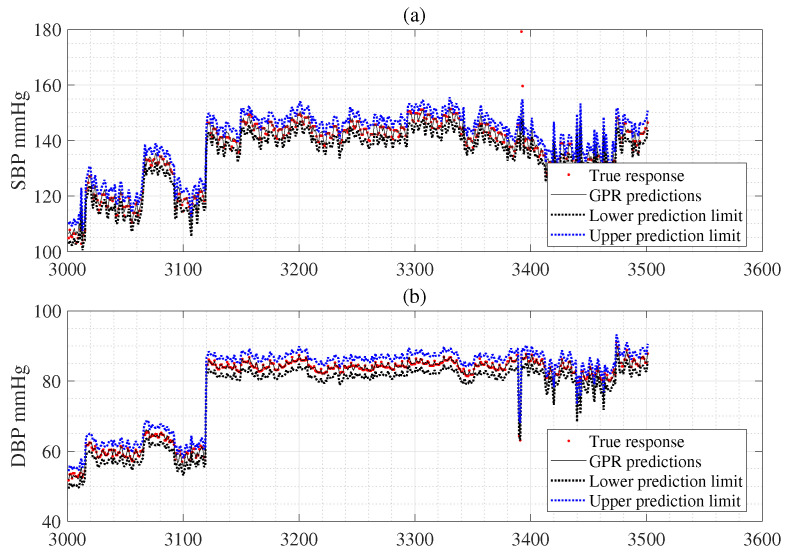
Panel (**a**) shows the CI estimation using the proposed WFD based on the combined GPR and GBA to represent the uncertainty of SBP estimation in cuff-less BP estimation (the 3000 to 3500 sample). Panel (**b**) shows the CI estimation using the proposed WFD based on the combined GPR and GBA to represent the uncertainty of DBP in cuff-less BP estimation (the 3000 to 3500 sample).

**Figure 6 bioengineering-12-00131-f006:**
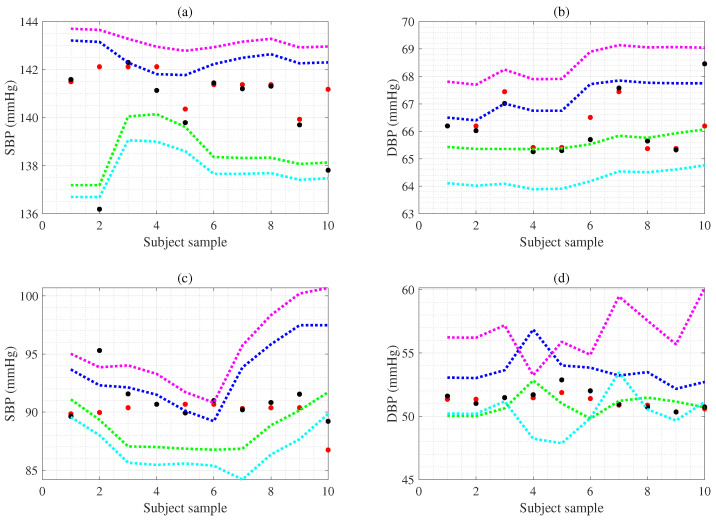
Panel (**a**) denotes the individual CI estimated using the bootstrap and uncertainty methods based on GPR, representing the uncertainty of SBP estimation from cuff-less BP estimates for one example subject (1 to 10 samples). Panel (**b**) denotes the individual CI estimated using the bootstrap and uncertainty methods based on GPR, representing the uncertainty of DBP estimation from cuff-less BP estimates for one example subject (1 to 10 samples). Panel (**c**) presents the CI individual estimated using the bootstrap and uncertainty methods based on GPR, representing the uncertainty of SBP estimation from cuff-less BP estimates for another example subject (1 to 10 samples). Panel (**d**) presents the individual CI estimated using the bootstrap and uncertainty methods based on GPR, representing the uncertainty of DBP estimation from cuff-less BP estimates for another example subject (1 to 10 samples).

**Table 1 bioengineering-12-00131-t001:** Features summary.

Features	Explanation
ST: Systolic time	Rise time from the PPG trough to the PPG peak, where the PPG peak is considered the systolic point of the ABP signal [10,31].
DT: Diastolic time	The time taken to fall from the current PPG waveform peak point to the next PPG waveform trough [10,31]
PIR: Pulse intensity ratio	Ratio of PPG peaks and troughs in the PPG [10,31]
HR: Heart rate	The reciprocal value of the time between consecutive ECG R-peaks [10,31]
PAT1: Pulse arrival time1	The time interval between the R-peak of the ECG waveform and the peak of the PPG waveform [31]
PAT2: Pulse arrival time2	The time interval between the R-peak of the ECG waveform and the maximum slope point (the first derivative peak value) of the PPG waveform [31]
PAT3: Pulse arrival time3	Width between R-peak of ECG and PPG trough [31]
LASI: Large artery stiffness index	The reciprocal value of the period from the peak of the PPG waveform to the inflection point closest to the diastolic trough [31]
AUI: Augmentation index	The ratio of the peak intensity of the PPG waveform to the intensity of the inflection point near the diastolic peak of the PPG,
	where this characteristic is considered a measure of the pressure wave reflection in the artery [31]
A1	Area under the PPG waveform curve from the diastolic trough to the maximum slope of the PPG waveform [31]
A2	Area under the curve from the point of maximum slope of the PPG waveform to the peak of the PPG waveform [31]
A3	Area under the curve from the PPG peak to the point of inflection closest to the diastolic peak
A4	Area under the curve from the inflection point of the PPG waveform to the diastolic trough of the next PPG waveform [31]
IPAR: Inflection point area ratio	ratio of S4/(S1 + S2 + S3) [31]
${\mathrm{PPG}}_{k}$	$\left(p_{m}-pd\right)/\left(p_{s}-pd\right)$
dPPG height (H)	PPG’s 1st derivative characteristics [10,31]
dPPG width (W)	PPG’s 1st derivative characteristics [10,31]
PH: ddPPG peak height	PPG’s 2nd derivative characteristics [10,31]
TH: ddPPG trough height (TH)	PPG’s 2nd derivative characteristics [10,31]
W: ddPPG width	PPG’s 2nd derivative characteristics [10,31]
H: ddPPG height	PPG’s 2nd derivative characteristics [10,31]
SA	Systolic area as Figure 3 [10]
DA	Diastolic area as Figure 3 [10]
areaPPG	Area under the envelope from the PPG
IBI	The first PPG’s systolic peak to the second PPG’s systolic peak
MXAP	Maximum amplitude of PPG waveform [32]
MIAP	Minimum amplitude of PPG waveform [32]
MEU	The blood’s viscosity [32]
FHR	The HR’s frequency [32]
CT:	Cycle time of one PPG waveform
MPSD:	Estimated max power spectral density (PSD)
MCORR:	Maximum correlation between PPG waveform and ABP waveform
Skew	PPG waveform skewness
Kurt	PPG waveform kurtosis
Entr	PPG waveform entropy
${\mathrm{SW}}_{10}$ and ${\mathrm{DW}}_{10}$:	the widths 10% of systolic and diastolic areas as Figure 3 [10]
${\mathrm{SW}}_{25}$ and ${\mathrm{DW}}_{25}$:	the widths 25% of systolic and diastolic areas as Figure 3
${\mathrm{SW}}_{33}$ and ${\mathrm{DW}}_{33}$:	the widths 33% of systolic and diastolic areas as Figure 3
${\mathrm{SW}}_{50}$ and ${\mathrm{DW}}_{50}$:	the widths 50% of systolic and diastolic areas as Figure 3
${\mathrm{SW}}_{66}$ and ${\mathrm{DW}}_{66}$:	the widths 66% of systolic and diastolic areas as Figure 3
${\mathrm{SW}}_{75}$ and ${\mathrm{DW}}_{75}$:	the widths 75% of systolic and diastolic areas as Figure 3

**Table 2 bioengineering-12-00131-t002:** Summary parameters for DPFE and WFD proposed based on conventional and GPR methods (based on training data).

Parameters Combined	SVM [38] DPFE	NN [10] DPFE	DNN [45] DPFE	RTree [39] DPFE	RF [42] DPFE	GPR [15] DPFE	GPR [15] DPFE&WFD
Number of sample	47,270	47,270	47,270	47,270	47,270	47,270	47,270
Number of validation sample	5908	5908	5908	5908	5908	5908	5908
Number of testing sample	5909	5909	5909	5909	5909	5909	5909
Number of feature	47	47	47	47	47	47	44
Output dimension	1	1	1	1	1	1	1
Learner				Tree	Tree		
Learning Cycle					30		
MaxNumSplits				47,269			
MinLeafSize				2	8		
KFold	10			10	10		
LayerSize		3					
Activations		ReLU					
Lambda		0.00001					
Standardize	1	1				1	1
Optimizer			Adam				
BoxConstraint	613						
Epsilon	0.2		1.0 × $10^{-8}$				
KernelScale	2.72						
KernelFunction	Gauss.					Exponential	Exponential
BasicFunction						None	None
NumHiddenUnits			500				
Layer			FeatureInputLayer				
			LstmLayer				
			FullyConnectedLayer				
MaxEpochs			500				
InitialLearnRate			0.005				
DropFactor			0.1				
Loss			mse				
SequenceLength			shortest				

**Table 3 bioengineering-12-00131-t003:** Algorithm complexity including validation and training of conventional and proposed methods on H/W (Intel^®^Core(TM) i5-9400 CPU 4.1 GHz, OS 64-bit, RAM 16.0 GB), and S/W (Matlab^®^2024 (The MathWorks Inc., Natick, Ma, USA) specifications).

Methods Combined	SVM [38] DPFE	NN [10] DPFE	DNN [45] DPFE	RTree [39] DPFE	RF [42] DPFE	GPR [15] DPFE	GPR [15] DPFE&WFD
Training time (s)	5217.4	41.8	10,875.4	48.1	128.1	14,533.6	15,576.2
Validation time (s)	8.5	2.5	3.3	1.8	2.6	4.30	4.41

**Table 4 bioengineering-12-00131-t004:** Reference ABP ranges in the dataset, where AAMI/ESH/ISO protocol defines the general population [36].

(mmHg)	Mean (mmHg)	SD (mmHg)	Mini (mmHg)	Max (mmHg)	≥160	≥140	≤100	≥100	≥85	≤60
SBP	133.1	24.8	71.2	199.7	15.6%	40.4%	9.2%			
DBP	69.3	14.6	49.9	149.5				3.8%	14.3%	32.9%
AAMI/ESH/ISO					5%	20%	5%	5%	20%	5%

**Table 5 bioengineering-12-00131-t005:** ME and SD relative to the reference ABP [36,48] for the conventional and proposed methods.

Method Combined (mmHg)	SVM [38] DPFE		NN [10] DPFE		DNN [45] DPFE		RTree [39] DPFE		RF [42] DPFE		GPR [15] DPFE		GPR [15] DPFE&WFD	
	SBP	DBP	SBP	DBP	SBP	DBP	SBP	DBP	SBP	DBP	SBP	DBP	SBP	DBP
ME	−0.16	−0.08	−1.05	−1.22	0.83	−0.07	−0.11	−0.02	−0.31	−0.10	−0.10	−0.03	−0.13	−0.04
SD	4.30	2.18	12.17	6.89	7.92	3.68	5.76	2.96	5.08	2.55	3.18	1.59	2.94	1.50

**Table 8 bioengineering-12-00131-t008:** Based on the BHS standard [49], where each result represents the average of 30 experimental data.

Method	Combined	SBP			DBP			SBP/DBP
		Mean Absolute Difference (%)			Mean Absolute Difference (%)			BHS
		≤5 mmHg	≤10 mmHg	≤15 mmHg	≤5 mmHg	≤10 mmHg	≤15 mmHg	Grade
SVM	DPFE	92.33	97.16	98.27	97.02	98.80	99.53	A/A
NN	DPFE	40.32	69.64	82.94	71.01	90.82	96.63	C/A
DNN	DPFE	51.90	81.71	93.64	86.74	97.98	99.46	B/A
RTree	DPFE	85.41	94.06	96.68	94.40	98.19	99.41	A/A
RF	DPFE	82.02	94.34	97.85	95.29	98.83	99.54	A/A
GPR	DPFE	93.62	97.95	99.10	98.15	99.59	99.86	A/A
GPR	DPFE&WFD	94.18	98.26	99.37	98.27	99.66	99.88	A/A
Grade A		60	85	95	60	85	95	[49]
Grade B		50	75	90	50	75	90	
Grade C		40	65	85	40	65	85	

**Table 9 bioengineering-12-00131-t009:** Comparision of uncertainties between the proposed and conventional methods, where n (=5909) denotes the number of test dataset, L and U denote the lower and upper limits, respectively, where (ex1.subject) denotes a subject and (ex2.subject) is another subject.

BP (mmHg) n (=)	SBP (SD) 95%CI	DBP (SD) 95%CI	SBP L (SD)	SBP U (SD)	DBP L (SD)	DBP U (SD)
${\mathrm{SVM}}_{\mathrm{Boot}}$	5.3 (7.1)	3.3 (3.9)	140.4 (18.5)	145.7 (18.0)	74.1 (10.4)	77.4 (10.5)
${\mathrm{RF}}_{\mathrm{Boot}}$	10.4 (7.2)	4.9 (4.1)	137.9 (18.5)	148.3 (17.7)	73.4 (10.7)	78.3 (10.9)
GPR	10.8 (7.3)	5.9 (4.0)	135.5 (20.1)	146.3 (20.3)	72.2 (12.6)	78.1 (12.5)
${\mathrm{GPR}}_{\mathrm{Boot}}$	4.7 (6.9)	3.0 (3.9)	140.7 (18.7)	145.4 (18.0)	74.3 (10.5)	77.3 (10.5)
${\mathrm{GPR}}_{\mathrm{Uncer}}$	8.9 (7.2)	6.4 (4.5)	138.6 (19.5)	147.5 (18.4)	72.6 (10.8)	79.0 (10.9)
${\mathrm{GPR}}_{\mathrm{Monte}}$	0.6 (0.9)	0.3 (0.8)	142.7 (18.2)	143.3 (18.1)	75.7 (10.4)	76.0 (10.5)
${\mathrm{GPR}}_{\mathrm{Boot}}$ (ex1.subject)	3.9 (1.5)	1.6 (0.4)	138.5 (1.1)	142.4 (0.5)	65.6 (0.3)	67.2 (0.6)
${\mathrm{GPR}}_{\mathrm{Uncer}}$ (ex1.subject)	5.4 (1.1)	4.2 (0.4)	137.8 (0.8)	143.2 (0.3)	64.3 (0.3)	68.5 (0.6)
${\mathrm{GPR}}_{\mathrm{Boot}}$ (ex2.subject)	4.8 (1.9)	2.7 (1.0)	88.6 (1.9)	93.4 (2.9)	50.9 (0.9)	53.6 (1.3)
${\mathrm{GPR}}_{\mathrm{Uncer}}$ (ex2.subject)	8.6 (2.9)	6.4 (1.3)	86.8 (1.9)	95.4 (3.4)	50.2 (1.6)	56.6 (2.1)

## Data Availability

The original contributions presented in this study are included in the article. Further inquiries can be directed to the corresponding author(s).

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
