# Peer review of "Improved Confidence-Interval Estimations Using Uncertainty Measure and Weighted Feature Decisions for Cuff-Less Blood-Pressure Measurements"

_bioengineering, 2025, doi:10.3390/bioengineering12020131_

Round 1

Reviewer 1 Report

Comments and Suggestions for Authors

The authors reported their work on the possible continuous blood pressure monitoring. It is a rather hot field in engineering, but not really needed for the management of hypertension. It is probably of some value for intensive critical care. There are some suggestions for revision of the manuscript.

1. The manuscript is very long, and hence can be shortened to some extent. 

2. The authors discussed clinical validation studies especially AAMI/ISO protocol. It is actually not really for the purpose of validation studies on cuffless devices. It is for cuff devices. The authors may consider the recently published ESH protocol for cuffless devices.

3. Some of the references are missing from the listing, but cited in the text. This reviewer tried but failed to find the explanation. If it is for purpose, the authors may need to provide explanations somewhere in the manuscript. 

Author Response

Response to the reviewers’ comments
“Individual Uncertainty Estimations Combining Dual-Stage Preprocessing with 
Feature Extraction in Continuous Blood Pressure Measurement ”
Soojeong Lee, Mugahed A. Al-antari, Gyanendra Prasad Joshi, 
Manuscript ID: bioengineering-3406702
General
We appreciate the valuable comments and suggestions of the reviewers on our paper very 
much. We have incorporated all the reviewers’ comments and suggestions in our submitted 
manuscript and given additional explanations. Our detailed responses are as follows.
Reviewer #1:
1. According to the comments, 
1. The manuscript is very long, and hence can be shortened to some extent.
1. Answer. 
We agree with the reviewer's comments, have reduced some parts, included some 
sentences.
2. According to the comments, 
The authors discussed clinical validation studies especially AAMI/ISO protocol. It is actually 
not really for the purpose of validation studies on cuffless devices. It is for cuff devices. The 
authors may consider the recently published ESH protocol for cuffless devices.
2. Answer. 
We validated our results and discussion using the ESH protocol published in 2023.
“The European Society of Hypertension (ESH) recommendations have validated cuffless BP measuring devices as the ME value is $ \leq $5 mmHg and the SD is $\leq$ 8 
mmHg [48].” 

3. According to the comments, 
Some of the references are missing from the listing, but cited in the text. This reviewer tried 
but failed to find the explanation. If it is for purpose, the authors may need to provide 
explanations somewhere in the manuscript.
3. Answer. We reviewed the references and checked the missing parts.

Reviewer 2 Report

Comments and Suggestions for Authors

The manuscript titled "Individual Uncertainty Estimations Combining Dual-Stage Preprocessing with Feature Extraction in Continuous Blood Pressure Measurement" presents a novel methodology that combines Gaussian Process Regression (GPR) with bootstrap techniques to estimate individualized confidence intervals (CIs) for continuous blood pressure (BP) monitoring. While the study is well-written and provides valuable insights, several key issues need to be addressed before it is considered for publication:

1.        The manuscript emphasizes the significance of addressing BP variability but does not examine physiological factors such as stress or activity levels that may impact signal quality and BP estimation accuracy.

2.        Although the preprocessing steps are designed to remove noise and outliers, the manuscript does not sufficiently discuss how artifacts caused by motion or sensor displacement, particularly in ECG signals, are managed. Additionally, the cited reference pertains only to PPG signal denoising, leaving a gap in explaining artifact handling for ECG signals. Examples or further clarification would be beneficial.

3.        The practical challenges of implementing the proposed methodology in wearable devices or clinical environments are not explored. Specifically, the computational demands of GPR could pose limitations for real-time applications, which warrants further discussion.

4.        The authors should provide more detail on how the derived confidence intervals (CIs) can be utilized in clinical decision-making or for monitoring patient health, bridging the gap between technical advancements and their practical utility.

5.        The manuscript does not discuss the integration of the proposed algorithm with existing BP monitoring devices. Addressing this would improve the study’s relevance to real-world applications and its potential for clinical adoption.

Author Response

Response to the reviewers’ comments
“Individual Uncertainty Estimations Combining Dual-Stage Preprocessing with 
Feature Extraction in Continuous Blood Pressure Measurement ”
Soojeong Lee, Mugahed A. Al-antari, Gyanendra Prasad Joshi, 
Manuscript ID: bioengineering-3406702
General
We appreciate the valuable comments and suggestions of the reviewers on our paper very 
much. We have incorporated all the reviewers’ comments and suggestions in our submitted 
manuscript and given additional explanations. Our detailed responses are as follows.
Reviewer #1:
1. According to the comments, 
1. The manuscript is very long, and hence can be shortened to some extent.
1. Answer. 
We agree with the reviewer's comments, have reduced some parts, included some 
sentences.
2. According to the comments, 
The authors discussed clinical validation studies especially AAMI/ISO protocol. It is actually 
not really for the purpose of validation studies on cuffless devices. It is for cuff devices. The 
authors may consider the recently published ESH protocol for cuffless devices.
2. Answer. 
We validated our results and discussion using the ESH protocol published in 2023.
“The European Society of Hypertension (ESH) recommendations have validated cuffless BP measuring devices as the ME value is $ \leq $5 mmHg and the SD is $\leq$ 8 
mmHg [48].”
Line 433-435.
3. According to the comments, 
Some of the references are missing from the listing, but cited in the text. This reviewer tried 
but failed to find the explanation. If it is for purpose, the authors may need to provide 
explanations somewhere in the manuscript.
3. Answer. We reviewed the references and checked the missing parts.
Reviewer #2:
1.According to the comments, 
The manuscript emphasizes the significance of addressing BP variability but does not 
examine physiological factors such as stress or activity levels that may impact signal quality 
and BP estimation accuracy.
1. Answer. We examined physiological factors such as stress or activity level and 
included the sentence as 
“This variability, influenced by stress, diet, exercise, climate, and disease, continuously 
fluctuates, making estimation more intricate.
Specifically, physical activity and psychological stress affected BP variability during the 
5 minutes before BP measurement. Psychological stress, expressed as negative emotions 
and workplace location, increased more significantly with physical activity than 
physical activity. Stressful situations were significantly associated with blood pressure 
elevation [1]. In addition, Schwartz et al. [2] reported that body position and location 
during measurement accounted for a significant portion of BP variability.” 
Line 23-30.
2. According to the comments, 
Although the preprocessing steps are designed to remove noise and outliers, the manuscript 
does not sufficiently discuss how artifacts caused by motion or sensor displacement, 
particularly in ECG signals, are managed. Additionally, the cited reference pertains only to 
PPG signal denoising, leaving a gap in explaining artifact handling for ECG signals. 
Examples or further clarification would be beneficial.
2.Answer. We included the artifacts in the ECG signal and added the cited reference of 
ECG signals as 
“Signal preprocessing removes outliers and extracts valuable features from PPG and 
ECG signals. In particular, various artifacts often contaminate ECG signals, which can 
degrade signal quality and hinder accurate analysis. Body movements or sensor 
displacements cause motion artifacts. These artifacts can cause irregularities in ECG 
waveforms, especially in walking or moving environments. Nagai et al. [21] introduced 
an algorithm to remove motion artifacts superimposed on ECG signals using stationary 
wavelet transform. Gunasekaran et al. [22] proposed a novel method to remove realtime artifacts from ECG signals using recursive independent component analysis (ICA). 
They describe a systematic preprocessing pipeline that adaptively estimates the mixing 
and demixing matrices of the ICA model while streaming data is being processed.”
“In our study, we first removed NaNs from the signals to maintain the alignment of 
each participant. We then performed a detailed validation using the signal quality index 
(SQI) algorithm to remove bad segment signals from the PPG and ECG signals, which 
is further described in the following subsections. Then, before denoising, we used an 
empirical Bayesian method using Cauchy to remove noise from the ECG and PPG data 
in low and high-frequency wavelets, discarded the last four detailed wavelet transform 
coefficients (d7, d8, d9, and d10), and approximated the coefficient a10 to remove 
artifacts from the signal [23].”
“Subsequently, we decomposed the PPG and ECG signals using a maximal overlap 
discrete wavelet transform (MODWT) with a Daubechies 8 wavelet (db8) [23]. We then 
reconstructed the PPG and ECG signals using an inverse MODWT without the highand low-frequency coefficients.”
Line 139-174.
3. According to the comments, 
. The practical challenges of implementing the proposed methodology in wearable devices or 
clinical environments are not explored. Specifically, the computational demands of GPR 
could pose limitations for real-time applications, which warrants further discussion.
3. Answer. We fully agreed with the reviewer’s comment and included the sentence 
concerning the computational demands of the GPR algorithm in wearable devices as 
“The computational requirements of Gaussian process regression (GPR) can pose realtime challenges for wearable devices, especially when dealing with large data sets or 
high-dimensional problems. Here, we describe how to address the computational cost of 
GPR, which will be the focus of our future research.
One powerful solution to these practical obstacles is sparse GPR. This approach allows 
us to approximate the posterior GP with a user-defined set of virtual training examples. 
This customization aspect will allow us to tailor the method to specific requirements by 
controlling the computational and memory complexity.
The GPR field has provided a wealth of sparse approximations to overcome 
computational limitations. Many authors expect to accurately process only a subset of 
latent variables while providing approximate but computationally cheaper processing 
for the remaining variables [53].
A straightforward way to address the computational complexity of large data sets is to 
use a subset of data methods. This approach selects a smaller subset (m < n) of 
observations from the total n and then applies the GPR model to these m points for 
estimation while ignoring the other (n - m) points. This small subset is called the 
active set or the inductive input set. When dealing with many observations, using exact 
methods for parameter estimation and making predictions on new data can be 
expensive (exact GPR). One of the approximate methods to solve this is the block 
coordinate descent (BCD) method [54], and we will apply this method in our study to 
analyze whether it is an efficient solution for the computational cost of GPR.”
Line 650-658.
4. According to the comments, 
The authors should provide more detail on how the derived confidence intervals (CIs) can 
be utilized in clinical decision-making or for monitoring patient health, bridging the gap 
between technical advancements and their practical utility.
4. Answer. We included detailed sentences concerning CIs in the introduction as 
“However, accurate BP measurement is a challenge due to the significant impact of the 
inherent physiological variability of BP. This variability, influenced by stress, diet, 
exercise, climate, and disease, continuously fluctuates, making estimation more 
intricate.
Specifically, physical activity and psychological stress affected BP variability during the 
5 minutes before BP measurement. Psychological stress, expressed as negative emotions 
and workplace location, increases more significantly with physical activity than 
psychological stress. Stressful situations were significantly associated with BP elevation 
[1]. In addition, Schwartz et al. [2] reported that body position and location during 
measurement accounted for a significant portion of BP variability.
Therefore, cuffless BP measurement devices require a method to measure the 
uncertainty that may arise from physiological variability. This is a significant limitation 
since most devices provide single-point estimates without confidence intervals (CIs).
Cuff-less BP measurements are naturally affected by various sources of uncertainty, 
leading to discrepancies between the measured value (the estimate) and the actual BP 
value (the reference BP) [3]. The sources of uncertainty can be categorized into random 
and systematic errors, which will be discussed in detail in the section on uncertainty 
measurement. When multiple sources of uncertainty influence a cuff-less BP 
measurement, the distribution of these BP measurements may converge to a Gaussian 
distribution as the number of uncertainties increases, regardless of the original 
distribution of the parameters representing these uncertainties [4]. 
Although several researchers in this field have attempted to study the uncertainty of bio 
signal measurements [5], an attempt has not been made to integrate the quality 
characteristics of the obtained signal and its compatibility with the estimation algorithm 
into an overall confidence measure of measurement accuracy.” 
“Therefore, it is necessary to provide CI for cuffless BP measurement to assess 
uncertainty, and it provides various estimated BP values that may include significant 
unknown factors [3]. If the integrated statistics show a wide range of CI, it can be 
considered a warning sign and alert patients, healthcare providers, and families to the 
risk.
However, CI estimation in cuffless BP measurement is still in the early stages of 
research and has not been actively studied. In addition, estimating the CI for each 
patient in practice requires multiple repeated BP measurements, even with a cuffless BP 
monitor. 
Unfortunately, measuring BP multiple times over an extended period for each patient 
using a cuff-less BP device is expensive and difficult to achieve because it does not 
ensure consistent conditions for reproducible readings [6]. 
Recently, Lee et al. [7] proposed a machine learning (ML) approach to simultaneously 
estimate BP and CI for cuffless BP measurements via hybrid feature selection based on 
photoplethysmography (PPG) with an electrocardiogram (ECG). However, this method 
results in a significant mean absolute error, which results in a CI that is too wide. If the 
CI is too broad, a warning may recommend discarding the measurement and starting a 
new one.
Line 22-59.
5. According to the comments, 
The manuscript does not discuss the integration of the proposed algorithm with existing BP 
monitoring devices. Addressing this would improve the study’s relevance to real-world 
applications and its potential for clinical adoption.
5. Answer. We included some sentences about integrating the proposed algorithm with 
existing BP monitoring devices.
“Our future research goal is to use the proposed algorithm in wearable devices. We 
perform cuff-free BP measurements locally on the patient's wearable device. The cufffree BP measurement device should be lightweight while providing accurate automated 
BP estimation and uncertainty measurements since wearable devices tend to have small 
and low-power processors. To improve the accuracy of the proposed algorithm, we will 
validate the proposed algorithm using a more significant number of data sets. We will 
also reduce the complexity of the proposed method so that it can be utilized in practical 
applications. We will discuss the proposed method's complexity reduction in detail 
below.”
Line 641-649.
Reviewer #3:
1.According to the comments, 
The attractive point is not so obvious. At this stage, these studies are of more theoretical than 
practical value. The Authors point out «Therefore, we cannot currently evaluate and use the 
quality characteristics of BP measurement results». If there is no way to check the obtained 
results with actual pressure measurements, then there is no practical importance yet. It is 
necessary to make appropriate explanations in the text of the article. Since in present article 
the Authors claim practical value.
1.Answer. We fixed these sentences and included explanations as 
“The purpose of measurement is to obtain the actual value of the measurand. The 
quantity to be measured is called the measurand [4]. 
We cannot know precisely how close the measured BP value is to the actual BP value. 
Therefore, BP estimates always contain uncertainty. The difference between the 
estimated and actual BP values is called the error, a well-known source of uncertainty. 
Here, uncertainty quantifies the doubt about the BP measurement result [20]. 
Considering the BP error, which consists of two parts, systematic error, and random 
error, we cannot know the BP error precisely because we do not know the actual BP 
value. Therefore, evaluating the BP measurement result without considering the 
uncertainty would be challenging. The quality and accuracy of the BP measurement 
result are characterized by its uncertainty, which defines the interval around the 
measured BP value within which we can judge with some probability that the true BP 
value exists.”
Line 309-319.
2.According to the comments, 
I would suggest adding sentences in article to discuss how that measurement could be relevant 
for real-life scenario.
2. Answer. 
“Our future research goal is to use the proposed algorithm in wearable devices. We 
perform cuff-free BP measurements locally on the patient's wearable device. The cufffree BP measurement device should be lightweight while providing accurate automated 
BP estimation and uncertainty measurements since wearable devices tend to have small 
and low-power processors. To improve the accuracy of the proposed algorithm, we will 
validate the proposed algorithm using a more significant number of data sets. We will 
also reduce the complexity of the proposed method so that it can be utilized in practical 
applications. We will discuss the proposed method's complexity reduction in detail 
below.”
“The computational requirements of Gaussian process regression (GPR) can pose realtime challenges for wearable devices, especially when dealing with large data sets or 
high-dimensional problems. Here, we describe how to address the computational cost of 
GPR, which will be the focus of our future research.
One powerful solution to these practical obstacles is sparse GPR. This approach allows 
us to approximate the posterior GP with a user-defined set of virtual training examples. 
This customization aspect will allow us to tailor the method to specific requirements by 
controlling the computational and memory complexity.
The GPR field has provided a wealth of sparse approximations to overcome 
computational limitations. Many authors expect to accurately process only a subset of 
latent variables while providing approximate but computationally cheaper processing 
for the remaining variables [53].
A straightforward way to address the computational complexity of large data sets is to 
use a subset of data methods. This approach selects a smaller subset (m < n) of 
observations from the total n and then applies the GPR model to these m points for 
estimation while ignoring the other (n - m) points. This small subset is called the 
active set or the inductive input set. When dealing with many observations, using exact 
methods for parameter estimation and making predictions on new data can be 
expensive (exact GPR). One of the approximate methods to solve this is the block 
coordinate descent method [54], and we will apply this method in our study to analyze 
whether it is an efficient solution for the computational cost of GPR.”
Line 641-668.
3.According to the comments, 
-“Reference BP” – how it is connected with the individual approach, because for each patient 
there will be an individual reference BP. Even this individual reference BP value will change 
due to age, the appearance of concomitant diseases, etc.. This approach may be used for heart 
disease prediction, but for measuring pressure in real life – problematic.
3. Answer. We agree with the reviewer's comment. Thus, we included the sentence as 
“Although the individual reference BP is linked to the reference BP based on the subject 
information provided by the MIMIC II data set. This study's extensive MIMIC II 
database records realistic physiological data of tens types of patients with noise or missing 
data gaps. Therefore, in future studies, we plan to use data from healthy people without 
BP disease when developing an algorithm for measuring BP in real life.”
Line 632-637.
4.According to the comments, 
-The abstract should have been rewritten. In the abstract, the authors should mention the main 
most interesting points of the review for the reader and the results obtained. Instead, the authors 
try to formulate the purpose and methods of the paper.
4. Answer. We modified the abstract as 
“This paper presented a method to improve confidence interval (CI) estimation using 
individual uncertainty measures and weighted feature decisions for cuff-less blood 
pressure (BP) measurement. We obtained uncertainty using Gaussian process regression 
(GPR). The CI obtained from the GPR model is computed using the distribution of BP 
estimates, which provides relatively wide CIs. Thus, we proposed a method to obtain 
improved CIs for individual subjects by applying bootstrap and uncertainty methods 
using the cuff-less BP estimates of each subject obtained through GPR. This study also 
introduced a novel method to estimate cuff-less BP with high fidelity by determining 
highly weighted features using weighted feature decisions. The standard deviation of 
the proposed method's mean error is 2.94 mmHg and 1.50 mmHg for systolic blood 
pressure (SBP) and (DBP), respectively. The mean absolute error results were obtained 
by weighted feature determination combining GPR and gradient boosting algorithms 
(GBA) for SBP (1.46 mmHg) and DBP (0.69 mmHg). The study confirmed that the BP 
estimates were within the CI based on the test samples of almost all subjects. The 
weighted feature decisions combining GPR and GBA were more accurate and reliable 
for cuff-less BP estimation.”
5.According to the comments, 
-The introduction part is too long and needs to be specifics.
5. Answer. We reduced some parts and re-wrote the introduction part
6.According to the comments, 
-I did not see the purpose of this paper. Methodologically in Introduction section there is no 
purpose but included some results. The Authors should note the subject of their research and 
expand this into the purpose of the paper, which is not there yet.
6. Answer. We re-wrote the introduction as 
“Our study aims to improve CI estimation using individual uncertainty measurement and 
weighted feature decisions (WFD) for cuff-less BP measurements. 
Thus, we use the Gaussian process regression (GPR) [15] to obtain an uncertainty as the 
ML algorithm. This uncertainty cannot be obtained directly from NN [16], SVM [17], or 
deep neural network (DNN) [18]. However, the CIs obtained from the GPR model are 
calculated based on the distribution of BP estimates, which provides a relatively wide CI. 
Although the probability that the BP estimate is included in relatively wide CIs increases, 
it has the disadvantage of reducing the reliability of the BP monitoring system. In addition, 
if the CI is too narrow, even a tiny change in the BP estimate can easily lead to an out-ofCI, which limits its role as a BP monitoring system. Securing an appropriate range of CIs 
is necessary to overcome these shortcomings.
“To address the above limitations, we propose a method to obtain improved CI for 
individual subjects by applying bootstrap [19] and uncertainty [20] methods based on the 
cuff-less BP estimates of each subject obtained through GPR. Second, this study 
introduces a novel methodology to estimate cuff-less BP with high fidelity by determining 
high-weighted features using weighted feature decisions (WFD) because weighted feature 
extraction is one of the fundamental steps in ML. 
The WFD method is an algorithm that automatically selects highly weighted features 
using a unified feature set. In the WFD methodology, we combine MRMR with the 
gradient boosting algorithm (GBA) to determine the feature set for the best BP estimation 
results. The role of WFD is to select the weighted feature subset with the smallest mean 
square error using GBA.”
Line 75~
7.According to the comments, 
-Authors wrote “The proposed method provides more accurate prediction performance and 
uncertainty by providing lower standard deviation (SD), mean absolute error, and CI”. What 
about false results?
7. Answer. We included some sentences as 
“The proposed method providing lower SD, MAE, and CI, it suggests that the method 
generally performs better in terms of accuracy and consistency. However, there are still 
situations where "false results" can occur, such as bias. 
If the proposed method has an inherent bias, even with low SD or MAE, it might 
consistently make errors in a specific direction (overestimating or underestimating), 
leading to false results. Also, suppose the proposed method is overfitted to the training 
data. In that case, it may produce accurate predictions on that specific data but fail to 
generalize well to unseen data, potentially leading to false results on new inputs.
A narrow CI might suggest high confidence in predictions, but if the model is poorly 
specified or the assumptions are wrong, this could still lead to false or inaccurate results 
despite low SD and MAE.”
Line 619-628.
8.According to the comments, 
-Authors wrote “Consequently, BP can be indirectly calculated by assuming that the PTT wave 
velocity is inversely proportional to the systolic blood pressure” – please explain how it will 
work in case of emergency pressure measurement.
8. Answer. We included the sentences about the PTT as 
“The relationship between PTT and SBP is based on the assumption that as SBP increases, 
arteries become stiffer (especially in hypertension). Thus, the wave velocity increases, 
shortening the PTT. Conversely, as BP decreases and the wave velocity decreases, the 
arterial wall becomes more flexible, lengthening the PTT. If PTT could be measured using 
pulse waves based on PPG signals in a wearable device, the time it takes for the pressure 
wave to travel between two points could be continuously recorded. Therefore, PTT could 
be measured in an emergency, and the inverse proportionality between PTT and BP could 
be used to estimate SBP.”
Line 234-241.
9.According to the comments, 
- Describing of the error calculation is missing taking into account the actual pressure values.
9. Answer. We included the error calculation as 
“ The MEs were calculated ($ME= \frac{1}{n}\sum_{i=1}^{n} me_{i}$) as $me_{i} 
= (ep_{i}-rp_{i})$ for each record $i$, where $ep$ represents the estimated BP (SBP or 
DBP), and $rp$ represents the reference BP. The MAEs were computed as ($MAE= 
\frac{1}{n}\sum_{i=1}^{n} |me_{i}|$). The SDs were calculated as ($\sqrt{\frac{1}{n-1} 
\sum_{i=1}^{n} (ME - me_{i})^2} $).” 
Line 438-441.
10.According to the comments, 
- What is the advantage of this method compared to those already in use in practice?
10. Answer. We included the advantages of our method in practice as 
“Conventional cuff-less BP monitoring devices only provide estimates at a single point 
without a CI representing uncertainty. These devices imply their inability to distinguish 
statistical variations from variations due to physiological processes \cite{soo1}. If this new 
method is used for cuff-less BP monitoring devices, wide CIs could trigger alarms, 
alerting the nurse station or primary care physician about potential patient risks in a 
home-based monitoring setting \cite{soo1}. Hence, predicting the CI for cuff-less BP 
monitoring is crucial for improving reliability. This study is essential because it is the first 
to predict individual uncertainty using the proposed DPFE and WFD method, which 
combines GPR and GBA, showing that almost BP estimates fall within the CIs based on 
test samples from all subjects, in Fig. \ref{fig5}. Thus, Figs. \ref{fig5} have substantial 
fluctuations.” 
Line 554-563.
“~The overall results reveal that the proposed DPFE and WFD, combining GPR and 
GBA, are more accurate and highly reliable for cuff-less BP estimation. This reliability 
should instill confidence and reassurance among healthcare professionals, researchers, 
and developers in health monitoring systems. The proposed methodology can 
continuously monitor BP changes using the estimated CI to estimate the uncertainty of 
cuff-less BP and hypertension risk.”
Line 606-618.
11.According to the comments, 
- Where did the pressure measurement data in section 6.1. Limitations come from?
11. Answer. We missed the data section and added the data section as 
“We utilized the multi-parameter intelligent monitoring (MIMIC-II) databases 
Goldberger \emph{et al. }\cite{Goldberger} at the ML storage center of the university of 
california, irvine. We acquired 3,000 records (participants) of ECG, finger PPG, and 
arterial BP signals at a sampling frequency of 125 Hz, sufficient to extract consecutive BP 
data. We obtained reference systolic blood pressure (SBP) and diastolic blood pressure 
(DBP) from the arterial BP signals. PPG and ECG signal waveforms were combined to 
obtain a feature set. We extracted statistical features from the PPG signal waveform and 
PPG signal frequency domain. The duration of the records in the database varied from 8 
seconds (s) to over 480 seconds (s). For consistency, a 20 s segment was extracted after 60 
s, resulting in 2500 samples from each record, enhancing the reliability of the obtained 
patient records. ~~”
Line 117-136.
12.According to the comments, 
-Please improve the quality of the Figs. Figs 2 and 4 are poor quality. Improve them. Edit the 
text part of the figures, increase the font size.
12. Answer. We modified the quality of the Figs. 2 and 4.

Reviewer 3 Report

Comments and Suggestions for Authors

Authors proposed a novel method to continuously obtain blood pressure values and estimate individual confidence intervals with high reliability using weighted feature selection based on Gaussian process regression. Manuscript is interesting but the paper has some problems.

Specific comments:

- The attractive point is not so obvious. At this stage, these studies are of more theoretical than practical value. The Authors point out «Therefore, we cannot currently evaluate and use the quality characteristics of BP measurement results».  If there is no way to check the obtained results with actual pressure measurements, then there is no practical importance yet. It is necessary to make appropriate explanations in the text of the article. Since in present article the Authors claim practical value.

- I would suggest adding sentences in article to discuss how that measurement could be relevant for real-life scenario.

-“Reference BP” – how it is connected with the individual approach, because for each patient there will be an individual reference BP. Even this individual reference BP value will change due to age, the appearance of concomitant diseases, etc.. This approach may be used for heart disease prediction, but for measuring pressure in real life – problematic.

-The abstract should have been rewritten. In the abstract, the authors should mention the main most interesting points of the review for the reader and the results obtained. Instead, the authors try to formulate the purpose and methods of the paper.

-The introduction part is too long and needs to be specifics.

-I did not see the purpose of this paper. Methodologically in Introduction section there is no purpose but included some results. The Authors should note the subject of their research and expand this into the purpose of the paper, which is not there yet.

-Authors wrote “The proposed method provides more accurate prediction performance and uncertainty by providing lower standard deviation (SD), mean absolute error, and CI”.  What about false results?

-Authors wrote “Consequently, BP can be indirectly calculated by assuming that the PTT wave velocity is inversely proportional to the systolic blood pressure” – please explain how it will work in case of emergency pressure measurement.

- Describing of the error calculation is missing taking into account the actual pressure values.

- What is the advantage of this method compared to those already in use in practice?

- Where did the pressure measurement data in section 6.1. Limitations come from?

-Please improve the quality of the Figs. Figs 2 and 4 are poor quality. Improve them. Edit the text part of the figures, increase the font size.

Author Response

(The authors gave the same response as above.)

Round 2

Reviewer 3 Report

Comments and Suggestions for Authors

Since the authors have addressed all the comments, I recommend its publication after minor revision.

Table 1. Features summary is too hard to read (too small print). Please correct it. 

On the pages 10 and 21 - a lot of extra space. Please correct it.  

Author Response

Now that we have fixed Table 1, we know and have experience that the paper editors help with table adjustment and space arrangement issues during the paper editing period.  Thank you for your review.